# Searching large-scale scRNA-seq databases via unbiased cell embedding with Cell BLAST

Zhi-Jie Cao [1,2], Lin Wei[1,2], Shen Lu[1], De-Chang Yang[1] & Ge Gao [1✉]

Single-cell RNA-seq (scRNA-seq) is being used widely to resolve cellular heterogeneity. With the rapid accumulation of public scRNA-seq data, an effective and efficient cell-querying method is critical for the utilization of the existing annotations to curate newly sequenced cells. Such a querying method should be based on an accurate cell-to-cell similarity measure, and capable of handling batch effects properly. Herein, we present Cell BLAST, an accurate and robust cell-querying method built on a neural network-based generative model and a customized cell-to-cell similarity metric. Through extensive benchmarks and case studies, we demonstrate the effectiveness of Cell BLAST in annotating discrete cell types and continuous cell differentiation potential, as well as identifying novel cell types. Powered by a well-curated reference database and a user-friendly Web server, Cell BLAST provides the one-stop solution for real-world scRNA-seq cell querying and annotation.

[1] Biomedical Pioneering Innovation Center (BIOPIC), Beijing Advanced Innovation Center for Genomics (ICG), Center for Bioinformatics (CBI), and State Key Laboratory of Protein and Plant Gene Research at School of Life Sciences, Peking University, 100871 Beijing, China. [2] These authors contributed equally: Zhi-Jie Cao, Lin Wei. ✉email: gaog@mail.cbi.pku.edu.cn

Technological advances during the past decade have led to rapid accumulation of large-scale single-cell RNA sequencing (scRNA-seq) data. Analogous to biological sequence analysis[1], existing annotations such as cell type and cell-differentiation potential in curated references can be utilized to annotate newly sequenced cells via a cell-querying algorithm, which unsupervisedly searches reference data for similar cells based on the transcriptome. Tools have been developed to achieve this using approximate cosine distance[2] or locality-sensitive hashing (LSH) Hamming distance[3,4] calculated from a subset of carefully selected genes. While these metrics per se are intuitive and computationally efficient, they may not reflect cell-to-cell similarity faithfully, and suffer from nonbiological confounding variation across datasets like batch effect[5,6]. Moreover, these methods do not come with a large-scale reference database with unified and comparable annotations, so users often need to search for proper reference data by themselves.

To address these challenges, we introduce Cell BLAST, a cell-querying tool employing a customized neural network-based generative model that effectively handles batch effect, as well as a cell-to-cell similarity metric specifically designed for the model (Fig. 1). We evaluate the query-based cell typing performance of Cell BLAST with extensive benchmark experiments, and use two case studies to demonstrate that Cell BLAST can further be utilized to predict continuous cell-differentiation potential and identify novel cell types. Finally, we provide a well-curated multispecies single-cell transcriptomics database (Animal Cell Atlas, ACA) and an easy-to-use Web interface for convenient exploratory analysis of hit cells.

## Results

**The Cell BLAST algorithm.** Cell BLAST uses a neural network-based generative model to adaptively learn a nonlinear projection from the high-dimensional transcriptomic space to a low-dimensional cell embedding space in an unsupervised manner using reference single-cell transcriptomes, with intra-reference batch effect corrected by adversarial alignment ("Methods", also see Fig. 1a for an illustrative diagram on the structure of the generative model used by Cell BLAST). When presented with

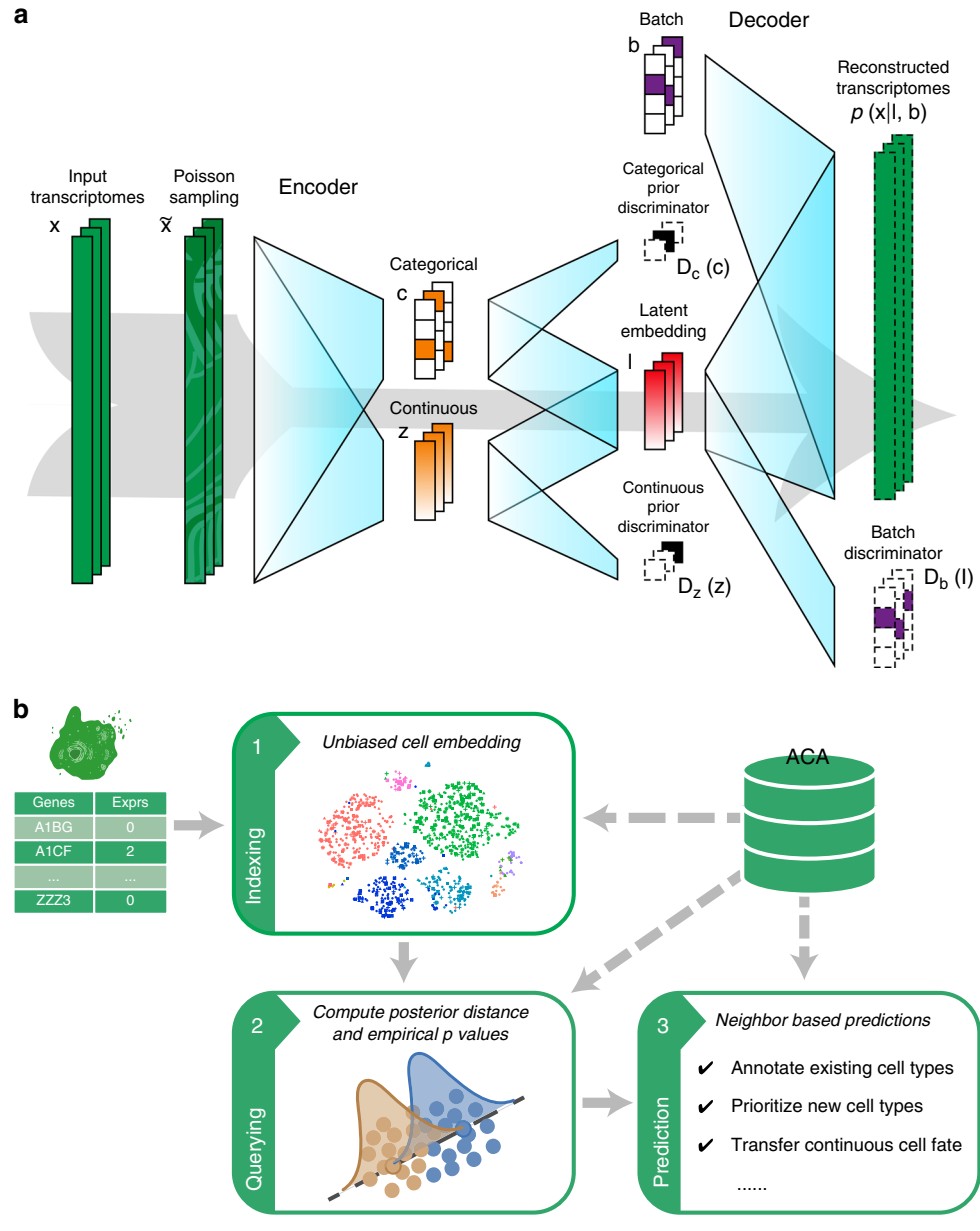

**Fig. 1 Cell BLAST model and workflow. a** Structure of the generative model used by Cell BLAST. **b** Overall Cell BLAST workflow.

query data, Cell BLAST uses the pretrained model to project individual query cells into the same low-dimensional space, and then utilizes the posterior distribution in the low-dimensional space, which characterizes the uncertainty of cell embeddings, to estimate cell-to-cell similarity precisely (Fig. 1b). Such a design also enables a special online-tuning mode that adaptively updates the model during querying to handle batch effect between query and reference data (e.g., in cross-species querying).

Reference cells with high similarity are then returned as query hits, the existing annotation of which can further be utilized to inform the annotation of query cells. Importantly, query cells with no significant hits are identified as unseen rather than incorrectly labeled, thus providing the opportunity to identify novel cell types.

**Evaluating cell-type resolution and batch-effect correction**. The effectiveness of our querying method relies on accurate dimension reduction and batch-effect correction by the generative model, so we first benchmarked the model against popular methods in each of these fields respectively (Supplementary Table 1) using real-world datasets (Supplementary Table 2). In the dimension-reduction benchmark, our model as well as DCA[7], PCA, and tSNE[8] are among the best performing ones in terms of cell-type resolution measured by mean average precision (MAP) (Supplementary Fig. 3).

For benchmarking batch-effect correction, we used combinations of datasets with overlapping cell types profiled, and treated each dataset as a batch. Our model achieves significantly better dataset mixing while maintaining high cell-type resolution at the same time (Fig. 2a). Cell-embedding visualization also validates that Cell BLAST effectively removed batch effect without blurring biological signals for datasets with considerable cell-type composition difference (Supplementary Fig. 4, Supplementary Data 1).

Notably, we found that the correction of inter-dataset batch effect does not automatically generalize to that within each dataset, which is most evident in the pancreatic datasets: while the canonical batch correction strategy works for cross-dataset batch effect (i.e., the batch effect observed among six independent datasets, Supplementary Fig. 4d), it does not handle within-dataset batch effects correctly (i.e., different donors in each dataset remain separated, Supplementary Fig. 5a–c). For such complex scenarios, our adversarial batch alignment strategy easily extends to correcting multiple levels of batch effect independently and simultaneously (Supplementary Fig. 5d–h).

**A posterior-based cell-to-cell similarity metric**. A proper cell-to-cell similarity metric in the embedding space should model the inputs' semantics accurately, and is critical for reliable cell querying. Common distance metrics like the Euclidean distance, though computationally efficient, may lead to unwanted artifacts (Supplementary Fig. 6a–d). Intuitively, low-dimensional embeddings of cells from the same cell type should reconstruct each other better than cells from different cell types. Thus, the posterior density, which models the uncertainty in cell embeddings ("Methods"), is expected to be flatter along the direction of the cell cluster, reflecting the local structure of data manifold. We reason that similarity metric could benefit from the additional information encoded in the posterior distribution of our generative model.

Employing a dedicated adversarial component[9], our model learns the genuine posterior distribution from data directly (in contrast to the fixed diagonal-covariance Gaussian used by methods based on canonical variational autoencoders[10], e.g., scVI[11]). Two-dimensional visualization of the "Baron_human"[12] and "Adam"[13] dataset illustrates that the posterior distribution of

our model captures the local structure of data manifold (Supplementary Fig. 6e, f), while the canonical model does not (Supplementary Fig. 7e, f).

Inspired by the Mahalanobis distance[14], we designed a custom distance metric (normalized projection distance, NPD) to quantify cell-to-cell similarity based on posterior distributions ("Methods", also see Supplementary Fig. 8a for an intuitive illustration). Distance metric ROC analysis in the two-dimensional embedding space shows that NPD is more accurate than Euclidean distance in distinguishing whether nearest reference cells are of the same type (Supplementary Fig. 6g, h). A similar analysis using ten-dimensional embedding space and independent query data also confirms the improvement in area under curve (AUC) with Cell BLAST variational posterior, while applying NPD with canonical Gaussian-based variational posterior did not increase AUC regardless of KL regularization weight (Supplementary Fig. 8c). To improve the comparability and interpretability of querying results, we compute an empirical P-value for each query hit as a measure of confidence during cell querying, by comparing the NPD to an empirical NULL distribution obtained from randomly selected pairs of cells in the reference data.

**Evaluating query-based cell typing**. We then evaluated the cell-querying performance against scmap[2] and CellFishing.jl[4] based on four groups of datasets (Supplementary Table 3), each including both positive queries (cell types existent in the reference) and negative queries (cell types non-existent in the reference). We compared the mean balanced accuracy (MBA) of each method in query-based cell-type prediction. An ideal cell-querying method should predict the correct cell types for positive queries, and reject negative queries at the same time (in which case "rejection" is the correct prediction). Cell BLAST achieves not only the highest overall MBA but also presents significantly superior specificity (i.e., higher negative type MBA) over others (Fig. 2b–c). Of note, the hits returned by Cell BLAST are more evenly distributed across multiple reference datasets (Supplementary Fig. 9e), further confirming better mitigation of batch effect.

Neural network-based models are often deemed as black boxes with no clear interpretation, impeding their adoption in high-stakes applications where understanding the reason behind model decisions is favorable. To help users' interpretation, we computed the gene expression gradient of the encoder neural network toward the position of each cell type in the embedding space ("Methods"). Larger gradient value for a gene means that cells highly expressing the gene will be embedded closer to the particular cell type, thus more likely to be predicted as the cell type during querying. To evaluate whether the gradients are reliable, we compared the gradient-based gene rankings with manually curated cell-type markers in the PanglaoDB database[15]. For the reference datasets used in the cell-querying benchmark, we found that genes with larger gradients for each cell type are significantly enriched for known markers of the particular cell type (Supplementary Fig. 10), suggesting that the internal logic of Cell BLAST models is generally consistent with prior knowledge.

Last but not least, we assessed scalability of these querying methods using the 1.3 M mouse brain dataset[16] subsampled to 1000–1,000,000 cells as reference. While scmap's querying time rises dramatically after 10,000 cells, Cell BLAST and CellFishing.jl scale well with increasing reference size (Fig. 2d).

**Rediscovery of a novel tracheal cell type**. The high specificity of Cell BLAST is critical for discovering novel cell types effectively. Two recent studies ("Montoro"[17] and "Plasschaert"[18]) independently

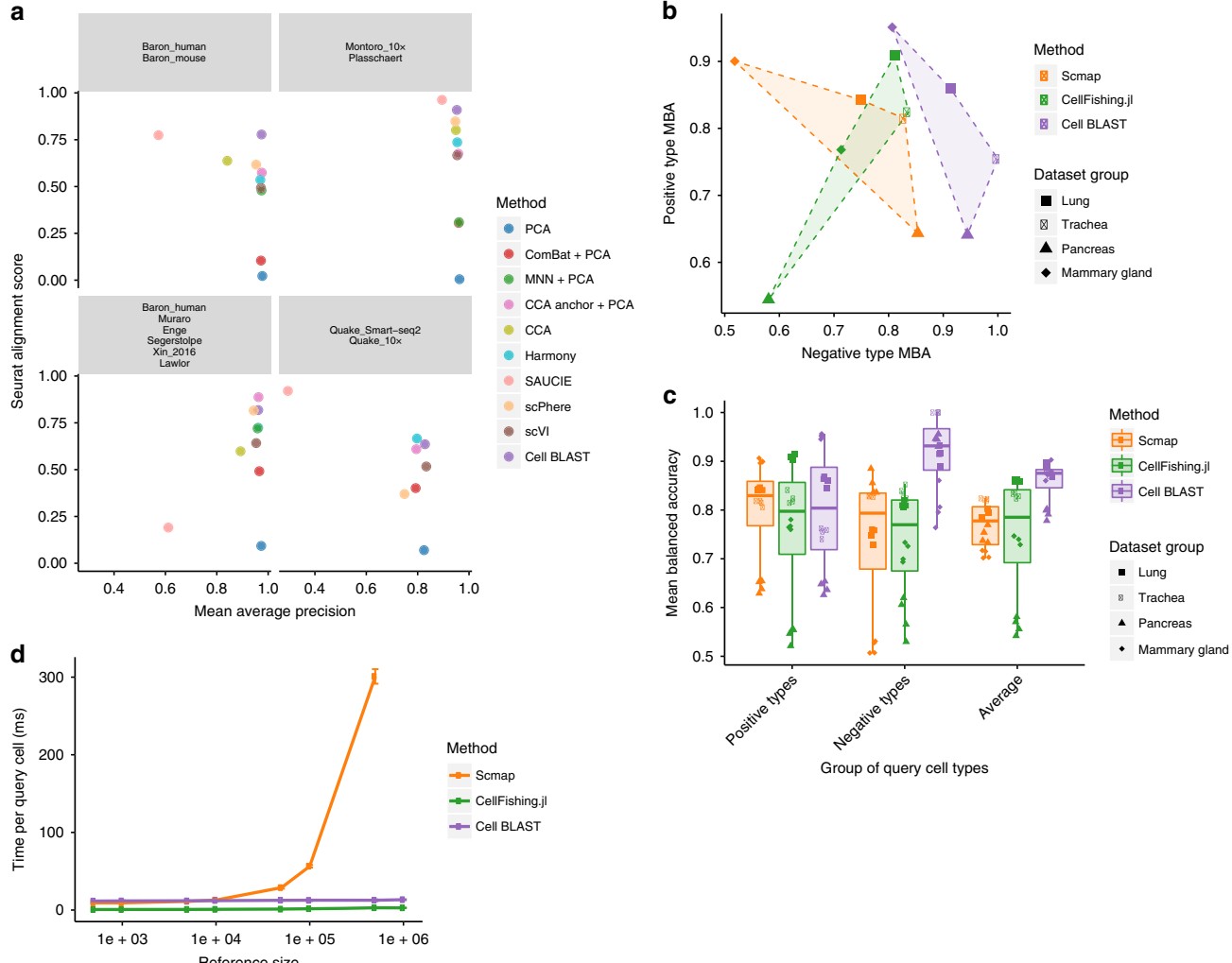

**Fig. 2 Cell BLAST benchmarking. a** Extent of dataset mixing as measured by Seurat alignment score, versus cell-type resolution, as measured by mean average precision, after batch effect correction in four groups of datasets. Both scores range between 0 and 1. Specifically, a high Seurat alignment score indicates that local neighborhoods consist of cells from different datasets uniformly rather than from the same dataset only, i.e., different datasets mix well. Meanwhile, mean average precision can be thought of as a generalization to nearest-neighbor accuracy, with larger values indicating higher cell-type resolution. It is reported to ensure that dataset mixing does not blur the true biological signal. CCA and MNN failed in the last dataset due to memory errors. **b** MBA of query-based cell typing on positive versus negative queries. Points of the same method are outlined for clarity. As CellFishing.jl does not come with a query-based prediction method, we used the same strategy as Cell BLAST, with Hamming distance = 120 as cutoff determined from grid searching for best balance between correctly predicting positive types and rejecting negative types across all four datasets (see "Methods" and Supplementary Fig. 8a, c for more details). **c** MBA of query-based cell typing on positive and negative queries as well as their arithmetic average ($n = 16$ experiments across four query groups for each method). Box plots indicate the median (center lines), 1st and 3rd quartiles (hinges), minimal and maximal point within 1.5 times the interquartile range starting from the hinges (whiskers). **d** Querying speed on reference datasets of different sizes subsampled from the 1.3 M mouse brain dataset[16] ($n = 4$ independent experiments for each method at each reference size). Error bars indicate mean ± s.d.

reported a rare tracheal cell type, pulmonary ionocyte. We artificially removed reported ionocytes from the "Montoro" dataset and used it as reference to annotate the "Plasschaert" dataset. In addition to achieving a high MBA of 0.873, Cell BLAST correctly rejected 14 of 19 "Plasschaert" ionocytes under the default cutoff (Fig. 3a), and highlights the existence of a putative novel cell type as a well-defined cluster (cluster 3) with large $P$-values (meaning higher confidence of rejection) among all 134 rejected cells (Fig. 3b–c). Further differential expression analysis shows that this is indeed an ionocyte cluster with highly expressed ionocyte markers including *Cftr*, *Foxi1*, and *Ascl3* (Supplementary Fig. 11).

As for other rejected cell clusters, we found that cluster 1, 2, 4, and 5 (Fig. 3b) are similar to cells of their originally annotated cell types (Supplementary Fig. 11, Supplementary Fig. 12a). Apart from the similarities, these rejected cells also exhibit different

transcriptional states from their matched counterparts revealed by GO enrichment analysis on differentially expressed genes (Supplementary Fig. 12b, c). More interestingly, we found that rejected cells in cluster 0 do not express markers of any retained cell type, but with a subgroup specifically expressing genes like *Cd52*, *Cd53*, *Itgb2* (Supplementary Fig. 11) related to immune response (Supplementary Fig. 12d). As an independent validation, we conducted principal component analysis (PCA) for each originally annotated cell type, and found that rejected cells and cells predicted as other cell types reside in a lower density region of the PC space (Supplementary Fig. 13), suggesting these cells are more or less atypical.

We tried the same analysis with other cell-querying methods, and found that scmap-cell[2] merely rejected 8 "Plasschaert" ionocytes (identified as cluster 4) out of all 319 rejections

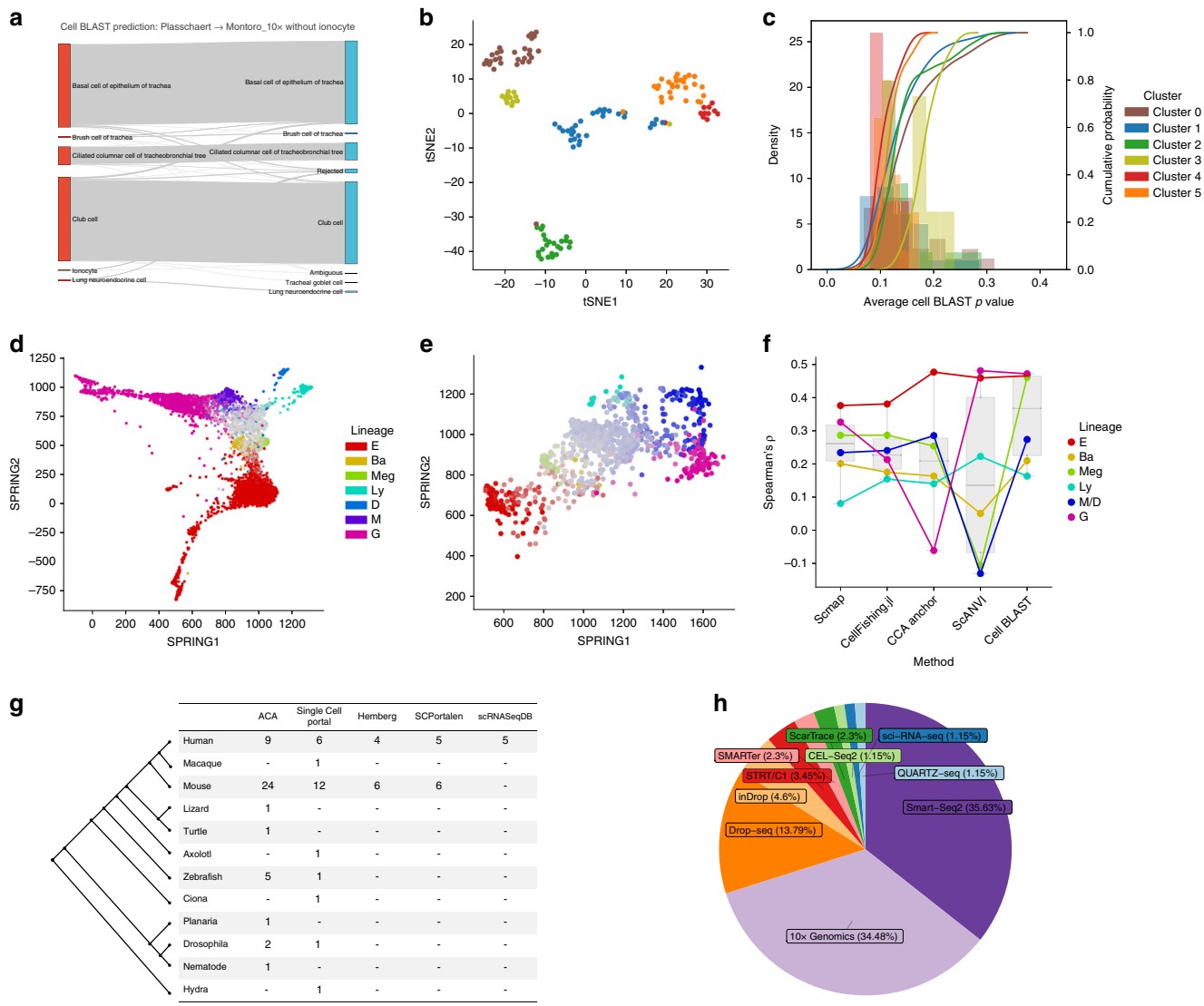

**Fig. 3 Cell BLAST application. a** Sankey plot comparing Cell BLAST predictions and original cell-type annotations for the "Plasschaert" dataset. **b** tSNE visualization of Cell BLAST-rejected cells, colored by unsupervised clustering. **c** Average Cell BLAST empirical *P*-value ("Methods") distribution of each cluster in (**b**). **d** SPRING visualization of cell embeddings learned on the "Tusi" dataset, colored by cell fate. Each of the seven terminal cell fates (E erythroid, Ba basophilic or mast, Meg megakaryocytic, Ly lymphocytic, D dendritic, M monocytic, G granulocytic neutrophil) is assigned a distinct color. The hue of each cell is then determined by the lineage with largest probability, while the saturation of each cell is determined by the entropy of cell fate distribution, i.e., terminally committed cells have more vibrant colors while undifferentiated cells appear to be gray. **e** SPRING visualization of the "Velten" dataset, colored by Cell BLAST predicted cell fate. **f** Spearman correlation between predicted cell fate probabilities and expression of known lineage markers in the "Velten" dataset ($n = 6$ lineages, where M and D are merged as they are indistinguishable in "Velten" according to the original publication). Box plots indicate the median (center lines), 1st and 3rd quartiles (hinges), minimum and maximum (whiskers). **g** Number of organs covered in each species for different single-cell transcriptomics databases, including the Single Cell Portal (https://singlecell.broadinstitute.org/single_cell), Hemberg collection[2], SCPortalen[23], and scRNASeqDB[24]. **h** Composition of different single-cell sequencing platforms in ACA.

(Supplementary Fig. 14a–c). Rejected cell clusters 0, 1, and 2 are similar to their originally annotated cell types. Cluster 3 is the same group of immune-related cells identified by Cell BLAST. Notably, lung neuroendocrine cells in rejected cluster 2 were assigned lower cosine similarity scores than ionocytes in rejected cluster 4 (Supplementary Fig. 14d, e), which is unreasonable. Finally, CellFishing.jl returned an excessive number of false rejections (Supplementary Fig. 14f). Among all methods, Cell BLAST achieved the highest ionocyte enrichment ratio in rejected cells (Supplementary Fig. 14g).

For ionocytes that are not rejected, we compared the prediction of scmap and Cell BLAST (Supplementary Fig. 15a). All five ionocytes predicted as club cells by Cell BLAST are also agreed on by scmap. They express higher levels of club cell markers like *Scgb3a2*, *Bpifa1*, *Sftpd*, and lower levels of ionocyte markers like *Asc3*, *Cfpr*, *Foxi1* compared with other ionocytes. With no indication of doublets based on total UMI (Unique Molecular Identifier) counts and detected gene numbers (Supplementary Fig. 15b, c), the result may suggest some intermediary cell state between club cells and ionocytes (but cross-contamination in the experimental procedures cannot be ruled out). Ionocytes predicted as other cell types by scmap, but rejected by Cell BLAST, all express high levels of ionocyte markers, but not markers of the alleged cell types (Supplementary Fig. 15a). These results also demonstrate that the querying result of Cell BLAST is more reliable.

**Prediction of continuous cell-differentiation potential.** Beyond cell typing, cell querying can also be used to infer continuous features. Our generative model combined with posterior-based similarity metric enables Cell BLAST to model the continuous spectrum of cell states more accurately. We demonstrate this using a study profiling mouse hematopoietic progenitor cells ("Tusi"[19]), in which the differentiation potential of each cell (i.e., cell fate) is characterized by its probability to differentiate into each of seven distinct lineages (i.e., cell fate probability, Fig. 3d, "Methods"). We first selected cells from one sequencing run as query and the other as reference to test whether continuous cell fate probabilities can be accurately transferred between experimental batches (Supplementary Fig. 16a). In addition to the cell-querying methods benchmarked above, we also incorporated two transfer learning methods recently developed for scRNA-seq data, i.e., CCA anchor[20] and scANVI[21]. Jensen–Shannon divergence between predicted cell fate probabilities and ground truth shows that Cell BLAST made the most accurate predictions (Supplementary Fig. 16b).

We further extended to inter-species annotation by trying to transfer cell fate annotation from the mouse "Tusi" dataset to an independent human hematopoietic progenitor dataset ("Velten"[22]) (Fig. 3e). Benefiting from its dedicated adversarial batch alignment-based online-tuning mode ("Methods"), Cell BLAST shows significantly higher correlation between the predicted cell fate probabilities and expression of known lineage markers for most lineages (Fig. 3f; also see Supplementary Fig. 17 for expression landscape of known lineage markers), while all other methods failed to properly handle the batch effect between species and produced biased predictions (Supplementary Fig. 16d–g).

**Constructing a large-scale well-curated reference database.** A comprehensive and well-curated reference database is crucial for the practical application of Cell BLAST. Based on public scRNA-seq datasets, we curated ACA, a high-quality reference database. To ensure a unified and high-resolution cell-type description, all records in ACA are collected and annotated using a standard procedure ("Methods"), with 98.9% of datasets manually curated with Cell Ontology, a structured controlled vocabulary for cell types. With 2,989,582 cells in total, ACA currently covers 27 distinct organs across eight species and five whole-organism atlases, offering a well-curated compendium for diverse species and organs compared to existing databases[2,23,24] (Fig. 3g–h; Supplementary Data 2). We trained our model on all ACA datasets. Notably, we found that the model works well in most cases with minimal hyperparameter tuning (cell-embedding visualizations, self-projection coverage and accuracy available on our website, Supplementary Fig. 18b).

A user-friendly Web server is publicly accessible at https://cblast.gao-lab.org, with all curated datasets and pretrained models available. Based on the wealth of resources, our website provides off-the-shelf querying service. Users can obtain querying hits and visualize cell-type predictions with minimal effort (Supplementary Fig. 18). For advanced users, a well-documented Python package implementing the Cell BLAST toolkit is also available, which enables model training on custom references and diverse downstream analyses.

## Discussion

By explicitly modeling multilevel batch effect as well as uncertainty in cell-to-cell similarity estimation, Cell BLAST is an accurate and robust querying algorithm for heterogeneous single-cell transcriptome datasets. Through extensive benchmark experiments under realistic settings, we compared Cell BLAST with relevant methods in the field of dimension reduction, batch-effect correction, and cell-querying, respectively, and achieved top or superior performance in most aspects.

The adversarial component in Cell BLAST enables a free form of variational posterior distribution, which could be learned from data directly. In contrast, the canonical variational autoencoder model used by scVI and several other tools[11,25–27] enforces Gaussian distribution (with diagonal-covariance matrices) for the variational posterior. Since the variational posterior is sufficiently accurate and efficient to sample from, the Cell BLAST model offers a unique opportunity of utilizing the posterior distribution for an improved, manifold-aware cell-to-cell similarity metric (NPD, see Supplementary Figs. 6–8 for more details). Two-dimensional visualizations revealed that the Cell BLAST variational posterior is more accurate than that of scVI (Supplementary Fig. 6–7), and that the improved variational posterior is crucial for the improvement in querying performance via posterior NPD (Supplementary Fig. 8c).

Since the generative model underlying Cell BLAST is trained in an unsupervised manner, Cell BLAST is not restricted to the task of cell typing like many classification-based cell-type annotation methods[28–30]. In fact, the same model can be applied to predict various types of features, including continuous cell-differentiation potential, as demonstrated in the case study of hematopoietic progenitor cells. We also showed that Cell BLAST reliably detects and highlights the existence of unseen query cells instead of reporting false positive predictions, which is crucial to the identification of novel cell types or aberrant cell states.

We noticed that while our adversarial batch alignment strategy can easily scale to a large number of batches (as the increase in parameter burden and computational cost is trivial when more batches are incorporated), integrating datasets in mega-scale may incur additional challenges like over-correcting batches with highly skewed cell-type composition. In addition, the Cell BLAST model, like all neural network-based models, is over-parameterized, and requires a reasonable number of samples to train properly. In our experiments, we found that a cell number of 3000 is generally sufficient for good performance, while small-sized datasets with fewer than 1000 cells may suffer from under-training issues.

Aiming to be a high-quality multispecies reference database, ACA is under regular update ("Methods"). With more and more scRNA-seq data generated recently, we, inspired by the classic Blast2GO[31] algorithm, implemented a query-based, ontology-aware inference strategy for assigning Cell Ontology terms during curation. Briefly, an unlabeled cell is first queried against existing ACA records, and the returned hit list is further processed to extract all Cell Ontology terms. For each identified Cell Ontology term, BLAST2CO calculates a hit-based confidence score, and then propagates the score to its ancestor terms. Finally, leaf terms (among those exceeding a given confidence threshold, by default 0.5) with maximal confidence score is assigned to the unlabeled cell as putative Cell Ontology curation ("Methods"; Supplementary Fig. 19a). Empirical case study shows that both the incorporation of hit similarity and ontology structure lead to improved Cell Ontology assignment, especially for cells marked as "ambiguous" by the standard majority voting strategy (Supplementary Fig. 19b–e). As more and more scRNA-seq data are being generated, we plan to continue maintaining the ACA database, to cover more species and organs, as well as to expand existing reference panels for increased generalizability.

In combination with a comprehensive, well-annotated database and an easy-to-use Web interface, Cell BLAST provides a one-stop solution for both bench biologists and bioinformaticians. We believe that as more scRNA-seq data become available, the utility of our toolkit will continue to improve over time.

## Methods

**The generative model.** In typical scRNA-seq experiments, single-cell gene expression profiles are in the form of $\mathbf{x}^{(i)} \in \mathbb{R}_{\geq 0}^{|\mathcal{G}|}$, $i = 1, 2, \ldots, N$, where $\mathcal{G}$ is the complete set of detected genes and $N$ is the total number of cells. To increase signal-to-noise ratio and reduce parameter space, a subset of informative genes $\mathcal{G}^*$ is selected using an established unsupervised gene selection method based on mean-variance relationship[32].

Attempting to capture biological variation of interest, e.g., discrete cell types and continuous cell states (but not technical factors, e.g., library size and batch effect, as discussed later), we model single-cell gene expression data as generated by an embedding variable $\mathbf{l}$ in $D$ dimensional latent space ($D \ll |\mathcal{G}^*|$) that follows a prior distribution $p(\mathbf{l})$[9]. Briefly, we model $\mathbf{l}$ as determined by two auxiliary latent variables of simple prior distributions:

$$\mathbf{l} = \mathbf{z} + \mathbf{Hc}$$
$$\mathbf{H} \in \mathbb{R}^{D \times K}, \mathbf{z} \in \mathbb{R}^D, \mathbf{z} \sim \mathrm{N}(\mathbf{0}, \mathbf{I}_D), \mathbf{c} \in \{0,1\}^K, \mathbf{c} \sim \mathrm{Cat}(K) \tag{1}$$

The continuous variable $\mathbf{z}$ models continuous variation in the transcriptome space, while the categorical variable $\mathbf{c}$ is set to model discrete cell clusters. Thus, the prior distribution for $\mathbf{l}$ is effectively a Gaussian mixture with component means parameterized by columns of the learnable matrix $\mathbf{H}$:

$$p(\mathbf{l}; \mathbf{H}) = \frac{1}{K} \sum_{k=1}^{K} \mathrm{N}\left(\mathbf{l}; \mathbf{H}_{.,k}, \mathbf{I}_D\right). \tag{2}$$

If the data contain only continuous variation of cell states and no discrete cell types, the categorical variable $\mathbf{c}$ can be left out and $\mathbf{l} = \mathbf{z}$ directly.

The generative process is implemented by a decoder neural network, denoted as Dec, which transforms $\mathbf{l}$ into parameters of the negative binomial distribution, which is a common choice for modeling UMI count-based scRNA-seq data[33–35]:

$$p(\mathbf{x}|\mathbf{l}; \phi_{\mathrm{Dec}}) = \mathrm{NB}\left(\mathbf{x}; \mathrm{Dec}_{\boldsymbol{\mu}}(\mathbf{l}, s; \phi_{\mathrm{Dec}}), \mathrm{Dec}_{\boldsymbol{\theta}}(\mathbf{l}; \phi_{\mathrm{Dec}})\right); \tag{3}$$

$$\mathrm{NB}(\mathbf{x}; \boldsymbol{\mu}, \boldsymbol{\theta}) = \prod_{j \in \mathcal{G}^*} \frac{\Gamma\left(\mathbf{x}_j + \boldsymbol{\theta}_j\right)}{\Gamma\left(\boldsymbol{\theta}_j\right)\Gamma\left(\mathbf{x}_j + 1\right)} \left(\frac{\boldsymbol{\mu}_j}{\boldsymbol{\theta}_j + \boldsymbol{\mu}_j}\right)^{\mathbf{x}_j} \left(\frac{\boldsymbol{\theta}_j}{\boldsymbol{\theta}_j + \boldsymbol{\mu}_j}\right)^{\boldsymbol{\theta}_j}, \tag{4}$$

where $\boldsymbol{\mu} \in \mathbb{R}_{\geq 0}^{|\mathcal{G}^*|}$ is the negative binomial mean, and $\boldsymbol{\theta} \in \mathbb{R}_{>0}^{|\mathcal{G}^*|}$ is the negative binomial dispersion. Each dimension of $\boldsymbol{\mu}$ and $\boldsymbol{\theta}$ corresponds to a gene in the selected subset $\mathcal{G}^*$. $\phi_{\mathrm{Dec}}$ denotes learnable parameters in the decoder network. $s$ is the size factor computed as the sum of expression values in the selected genes $s = \sum_{j \in \mathcal{G}^*} \mathbf{x}_j$. Variation in the size factor is primarily caused by technical factors, such as capture/amplification efficiency and sequencing depth[36]. Feeding $s$ to the decoder makes it possible to decouple such technical factors from the cell embedding, and concentrate the information learned in $\mathbf{l}$ on biological variations.

Apart from the negative binomial (NB) distribution, zero-inflated negative binomial (ZINB) is also widely adopted when modeling scRNA-seq data, motivated by the belief that scRNA-seq data contain excessive number of zero entries caused by dropout events[7,11,37]. Nevertheless, recent studies[38–40] have shown that most zero entries can actually be explained by biological variation and random sampling process, and that negative binomial is sufficient for modeling UMI-based scRNA-seq data. Further considering the identifiability issue of ZINB and risk of overfitting, we choose to employ the simpler NB as the generative distribution.

**Approximate posterior and model optimization.** For cell querying in the embedding space, we are most interested in the posterior distribution $p(\mathbf{l}|\mathbf{x}; \phi_{\mathrm{Dec}}, \mathbf{H})$ of cell embedding $\mathbf{l}$. It gives a probability density in the embedding space $\mathbb{R}^D$ that characterizes likely embedding positions of a cell based on its observed gene expression profile:

$$p(\mathbf{l}|\mathbf{x}; \phi_{\mathrm{Dec}}, \mathbf{H}) = \frac{p(\mathbf{x}|\mathbf{l}; \phi_{\mathrm{Dec}})p(\mathbf{l}; \mathbf{H})}{\int p(\mathbf{x}|\mathbf{l}; \phi_{\mathrm{Dec}})p(\mathbf{l}; \mathbf{H})d\mathbf{l}}. \tag{5}$$

Note that the uncertainty in cell embedding $\mathbf{l}$ originate from the uncertainty in generative distribution $p(\mathbf{x}|\mathbf{l}; \phi_{\mathrm{Dec}})$, which is designed to fit the noisy scRNA-seq data in the first place.

It is acknowledged that the observation $\mathbf{x}$ of a cell's gene expression profile obtained by scRNA-seq is not an exact depiction of its transcriptional state, but rather subject to both biological noise inherent in gene expression mechanisms, and detection noise due to technical limitations[33,41]. Thus, it is natural to represent cells in the embedding space by their posterior density, which captures the uncertainty.

Given the complexity of the generative model, direct maximum likelihood estimation and posterior inference are intractable. For efficient training and inference, a stochastic encoder neural network is introduced to approximate sampling from the actual posterior distribution, which we denote as $q(\mathbf{l}|\mathbf{x}; \phi_{\mathrm{Enc}}, \mathbf{H})$, where $\phi_{\mathrm{Enc}}$ denotes the learnable parameters in the encoder network. The stochastic encoder projects the gene expression profile $\mathbf{x}$ to the embedding $\mathbf{l}$ via the

following steps:

$$\hat{\mathbf{x}} = \frac{10^4 \cdot \mathbf{x}}{\sum_{j \in \mathcal{G}} \mathbf{x}_j}; \tag{6}$$

$$\tilde{\mathbf{x}} \sim \mathrm{Poisson}(\hat{\mathbf{x}}); \tag{7}$$

$$\mathbf{z} = \mathrm{Enc}_{\mathbf{z}}(\tilde{\mathbf{x}}; \phi_{\mathrm{Enc}}), \mathbf{c} = \mathrm{Enc}_{\mathbf{c}}(\tilde{\mathbf{x}}; \phi_{\mathrm{Enc}}); \tag{8}$$

$$\mathbf{l} = \mathbf{z} + \mathbf{Hc}. \tag{9}$$

Note that we first normalize the input transcriptome to the same scale of $10^4$, so that the encoder can generalize across datasets with vastly different sequencing depth. The normalization factor in Eq. (6) is computed using all genes $\mathcal{G}$, instead of the selected subset $\mathcal{G}^*$ to avoid skewing the data. Stochasticity in the encoding process is introduced by sampling from the Poisson distribution with rate = $\hat{\mathbf{x}}$. The choice of Poisson is arbitrary, and mainly serves as a source of randomness. The Poisson samples then go through the encoder network to give final samples of the approximate posterior. As a result, the approximate posterior is also parameterized by the encoder neural network, making it more flexible than conventional distribution families, like diagonal-variance Gaussian typically used in canonical variational autoencoders[10]. To obtain point estimates of the cell embeddings, we skip the Poisson sampling step and set $\tilde{\mathbf{x}} = \hat{\mathbf{x}}$ directly.

Training objectives for the model are:

$$\max_{\phi_{\mathrm{Enc}}, \phi_{\mathrm{Dec}}, \mathbf{H}} \mathbb{E}_{\mathbf{x} \sim p_{\mathrm{data}}(\mathbf{x})} \begin{bmatrix} \mathbb{E}_{\mathbf{l} \sim q(\mathbf{l}|\mathbf{x}; \phi_{\mathrm{Enc}}, \mathbf{H})} \log p(\mathbf{x}|\mathbf{l}; \phi_{\mathrm{Dec}}) \\ + \lambda_{\mathbf{z}} \cdot \mathbb{E}_{\mathbf{z} \sim q(\mathbf{z}|\mathbf{x}; \phi_{\mathrm{Enc}})} \log \mathrm{D}_{\mathbf{z}}(\mathbf{z}; \phi_{D_{\mathbf{z}}}) \\ + \lambda_{\mathbf{c}} \cdot \mathbb{E}_{\mathbf{c} \sim q(\mathbf{c}|\mathbf{x}; \phi_{\mathrm{Enc}})} \log \mathrm{D}_{\mathbf{c}}(\mathbf{c}; \phi_{D_{\mathbf{c}}}) \end{bmatrix}; \tag{10}$$

$$\max_{\phi_{D_{\mathbf{z}}}} \lambda_{\mathbf{z}} \cdot \left( \mathbb{E}_{\mathbf{z} \sim p(\mathbf{z})} \log \mathrm{D}_{\mathbf{z}}(\mathbf{z}; \phi_{D_{\mathbf{z}}}) + \mathbb{E}_{\mathbf{x} \sim p_{\mathrm{data}}(\mathbf{x})} \mathbb{E}_{\mathbf{z} \sim q(\mathbf{z}|\mathbf{x}; \phi_{\mathrm{Enc}})} \log\left(1 - \mathrm{D}_{\mathbf{z}}(\mathbf{z}; \phi_{D_{\mathbf{z}}})\right)\right); \tag{11}$$

$$\max_{\phi_{D_{\mathbf{c}}}} \lambda_{\mathbf{c}} \cdot \left( \mathbb{E}_{\mathbf{c} \sim p(\mathbf{c})} \log \mathrm{D}_{\mathbf{c}}(\mathbf{c}; \phi_{D_{\mathbf{c}}}) + \mathbb{E}_{\mathbf{x} \sim p_{\mathrm{data}}(\mathbf{x})} \mathbb{E}_{\mathbf{c} \sim q(\mathbf{c}|\mathbf{x}; \phi_{\mathrm{Enc}})} \log\left(1 - \mathrm{D}_{\mathbf{c}}(\mathbf{c}; \phi_{D_{\mathbf{c}}})\right)\right). \tag{12}$$

where samples from $q(\mathbf{l}|\mathbf{x}; \phi_{\mathrm{Enc}}, \mathbf{H})$ can be obtained from Eq. (9), and samples from $q(\mathbf{z}|\mathbf{x}; \phi_{\mathrm{Enc}})$ and $q(\mathbf{c}|\mathbf{x}; \phi_{\mathrm{Enc}})$ can be obtained from Eq. (8). $\mathrm{D}_{\mathbf{z}}$ and $\mathrm{D}_{\mathbf{c}}$ are discriminator neural networks for $\mathbf{z}$ and $\mathbf{c}$, respectively, which output the probability that a sample of the latent variable is from the prior rather than the posterior. $\phi_{D_{\mathbf{z}}}$ and $\phi_{D_{\mathbf{c}}}$ are learnable parameters in the discriminator networks. Eq. (10) maximizes an approximate lower bound of data likelihood, while the adversarial training between the encoder and discriminators effectively drives the encoded $\mathbf{z}$ and $\mathbf{c}$ to match prior distributions $p(\mathbf{z})$ and $p(\mathbf{c})$. $\lambda_{\mathbf{z}}$ and $\lambda_{\mathbf{c}}$ are hyperparameters controlling prior matching strength. Despite the use of adversarial training, the model is much easier and more stable to train than canonical GANs, because the adversarial component operates on low-dimensional spaces mapped from high-dimensional gene expression data (in contrary to conventional GANs, where the adversarial component operates on high-dimensional data generated from low-dimensional variables). It effectively extricates the model from the disjoint support problem, which is considered a major contributing factor to training instability of canonical GANs[42].

Stochastic gradient descent (SGD) with minibatches is applied to optimize the loss functions. Each SGD iteration is divided into two steps. In the first step, all discriminator networks (including $\mathrm{D}_{\mathbf{z}}$, $\mathrm{D}_{\mathbf{c}}$, and $\mathrm{D}_{\mathbf{b}}$ as described in the next section) are updated simultaneously. In the second step, we update the encoder and decoder networks to fit gene expression data and counteract the discriminator networks. This keeps different network components synchronized as much as possible. The RMSProp optimization algorithm with no momentum term is employed to ensure stability of adversarial training. Meanwhile, we also incorporated some finer architectural designs inspired by scVI[11], specifically the logarithm transformation before encoder input, and softmax output scaled by library size when computing $\boldsymbol{\mu}$. The model is implemented using the Tensorflow[43] Python library. Visualization of loss values during training reveals stable training dynamics and fast convergence (Supplementary Fig. 1a, b).

Model sensitivity to key hyperparameters is tested in Supplementary Fig. 1c. Specifically, we start from a default set of manually optimized hyperparameters (dimensionality = 10, hidden_layer = 128, depth = 1, cluster = 20, lambda_prior = 0.001, prob_module = "NB"), and then alter each hyperparameter with the others fixed. We found that the model is robust to hyperparameter settings within reasonable ranges, and the effect of hyperparameter adjustment on performance is similar across datasets.

A potential risk of choosing NB over ZINB is that the model may underfit data generated by plate-based protocols like Smart-seq2[44], which could indeed be zero-inflated, possibly due to the lack of UMI deduplication[40]. As such, we further compared NB vs ZINB models fitted on Smart-seq2 vs UMI-based data (Supplementary Fig. 2). The performance difference between NB and ZINB is generally small ($\Delta$MAP < 0.002 in 10 out of 11 datasets), though we indeed see that

ZINB performs slightly better than NB in all three Smart-seq2 datasets, while it performs slightly worse than NB in six out of eight UMI-based datasets, consistent with the speculation that plate-based non-UMI protocols are more zero-inflated than UMI-based protocols.

**Adversarial batch alignment**. As a natural extension to the adversarial prior matching strategy described in the previous section, and following recent works in domain adaptation[45–47], we propose an adversarial strategy to correct batch effect by aligning cells from different batches in the embedding space.

To perform adversarial batch alignment, we require the batch membership of each cell to be known, which is typically the case for real-world data. We denote the batch membership of each cell as $\mathbf{b} \in \{0,1\}^B$, where $B$ is the number of batches. If a cell belongs to the $i$th batch, then $\mathbf{b}_i = 1$ and $\mathbf{b}_j = 0, j \neq i$. Here, we write the categorical batch distribution $p(\mathbf{b})$ as:

$$p(\mathbf{b}_i = 1) = \mathbf{w}_i, \quad \sum_{i=1}^{B} \mathbf{w}_i = 1, \tag{13}$$

where $\mathbf{w}_i$ is the relative proportion of cells in the $i$th batch.

One problem with batch alignment is that, when batch information is successfully erased in $\mathbf{l}$, the decoder would struggle to fit the observed gene expression data $\mathbf{x}$, which is influenced by batch effect in the first place. So, the generative distribution is extended to condition on $\mathbf{b}$ as well:

$$p(\mathbf{x}|\mathbf{l}, \mathbf{b}; \phi_{\text{Dec}}) = \text{NB}\Big(\mathbf{x}; \text{Dec}_{\boldsymbol{\mu}}(\mathbf{l}, \mathbf{b}, s; \phi_{\text{Dec}}), \text{Dec}_{\boldsymbol{\theta}}(\mathbf{l}, \mathbf{b}; \phi_{\text{Dec}})\Big). \tag{14}$$

Model structure with the adversarial batch alignment component is also illustrated in Fig. 1a.

Adversarial batch alignment introduces an additional loss:

$$\max_{\phi_{\text{Enc}}, \phi_{\text{Dec}}, \mathbf{H}} \mathbb{E}_{\mathbf{b} \sim p(\mathbf{b})} \mathbb{E}_{\mathbf{x} \sim p(\mathbf{x}|\mathbf{b})} \mathbb{E}_{\mathbf{l} \sim q(\mathbf{l}|\mathbf{x}; \phi_{\text{Enc}}, \mathbf{H})} \Big[ \mathcal{L}_{\text{base}} - \lambda_{\mathbf{b}} \cdot \mathbf{b}^{\top} \log D_{\mathbf{b}}\big(\mathbf{l}; \phi_{D_{\mathbf{b}}}\big) \Big]; \tag{15}$$

$$\max_{\phi_{D_{\mathbf{b}}}} \mathbb{E}_{\mathbf{b} \sim p(\mathbf{b})} \mathbb{E}_{\mathbf{x} \sim p(\mathbf{x}|\mathbf{b})} \mathbb{E}_{\mathbf{l} \sim q(\mathbf{l}|\mathbf{x}; \phi_{\text{Enc}}, \mathbf{H})} \Big[ \lambda_{\mathbf{b}} \cdot \mathbf{b}^{\top} \log D_{\mathbf{b}}\big(\mathbf{l}; \phi_{D_{\mathbf{b}}}\big) \Big], \tag{16}$$

where $\mathcal{L}_{\text{base}}$ denotes the basic loss function in Eq. (10). $D_{\mathbf{b}}$ is a multiclass batch discriminator network of $B$ outputs. It is trained to predict the batch membership of cells based on their embeddings $\mathbf{l}$ (Eq. (16)). The encoder is trained in the opposite direction to fool the batch discriminator (Eq. (15)), effectively aligning different batches in the embedding space. $\lambda_{\mathbf{b}}$ is a hyperparameter controlling batch alignment strength.

Below, we extend the derivation in the original GAN paper[48] to demonstrate the effect of adversarial batch alignment in Eqs. (15) and (16). We first focus on the alignment term, ignoring the $\mathcal{L}_{\text{base}}$ term and scaling parameter $\lambda_{\mathbf{b}}$.

To simplify notation, we fuse the data distribution $\mathbf{x} \sim p(\mathbf{x}|\mathbf{b})$ and encoder transformation $\mathbf{l} \sim q(\mathbf{l}|\mathbf{x}; \phi_{\text{Enc}}, \mathbf{H})$ into $q(\mathbf{l}|\mathbf{b}; \phi_{\text{Enc}}, \mathbf{H})$. The distribution $q(\mathbf{l}|\mathbf{b}=i; \phi_{\text{Enc}}, \mathbf{H})$ thus represents the cell-embedding distribution of the $i$th batch. To further reduce cluttering, we neglect the parameters $\phi_{\text{Enc}}, \mathbf{H}$ and $\phi_{D_{\mathbf{b}}}$, but be aware that $q(\mathbf{l}|\mathbf{b}=i)$ is controlled by $\phi_{\text{Enc}}$ and $\mathbf{H}$, while $D_{\mathbf{b}}(\mathbf{l})$ is controlled by $\phi_{D_{\mathbf{b}}}$. We may then rewrite Eqs. (15), (16) as:

$$\min_{\phi_{\text{Enc}}, \mathbf{H}} \sum_{i=1}^{B} \mathbf{w}_i \mathbb{E}_{\mathbf{l} \sim q(\mathbf{l}|\mathbf{b}=i)} \log D_{\mathbf{b}_i}(\mathbf{l}); \tag{17}$$

$$\max_{\phi_{D_{\mathbf{b}}}} \sum_{i=1}^{B} \mathbf{w}_i \mathbb{E}_{\mathbf{l} \sim q(\mathbf{l}|\mathbf{b}=i)} \log D_{\mathbf{b}_i}(\mathbf{l}). \tag{18}$$

Here $D_{\mathbf{b}_i}(\mathbf{l})$ denotes the $i$th dimension of the discriminator output, which predicts the probability of a cell belonging to the $i$th batch. We assume $D_{\mathbf{b}}$ to have sufficient capacity to approach the optimum, which is generally reasonable in the case of neural networks. The global optimum of Eq. (18) is reached when $D_{\mathbf{b}}$ outputs optimal batch membership probabilities at every $\mathbf{l}$:

$$\max_{D_{\mathbf{b}_i}(\mathbf{l})} \sum_{i=1}^{B} \mathbf{w}_i q(\mathbf{l}|\mathbf{b}=i) \log D_{\mathbf{b}_i}(\mathbf{l}), \quad \text{s.t.} \sum_{i=1}^{B} D_{\mathbf{b}_i}(\mathbf{l}) = 1. \tag{19}$$

The solution to the above maximization is given by:

$$D_{\mathbf{b}_i}^{*}(\mathbf{l}) = \frac{\mathbf{w}_i q(\mathbf{l}|\mathbf{b}=i)}{\sum_{i=1}^{B} \mathbf{w}_i q(\mathbf{l}|\mathbf{b}=i)} \tag{20}$$

i.e., the $i$th output of optimal batch discriminator is the relative cell-embedding

density of the $i$th batch. Substituting $D_{\mathbf{b}}^{*}(\mathbf{l})$ back into Eq. (17), we obtain:

$$\begin{aligned} &\sum_{i=1}^{B} \mathbf{w}_i \mathbb{E}_{\mathbf{l} \sim q(\mathbf{l}|\mathbf{b}=i)} \log \frac{\mathbf{w}_i q(\mathbf{l}|\mathbf{b}=i)}{\sum_{i=1}^{B} \mathbf{w}_i q(\mathbf{l}|\mathbf{b}=i)} \\ &= \sum_{i=1}^{B} \mathbf{w}_i \mathbb{E}_{\mathbf{l} \sim q(\mathbf{l}|\mathbf{b}=i)} \log \frac{q(\mathbf{l}|\mathbf{b}=i)}{\sum_{i=1}^{B} \mathbf{w}_i q(\mathbf{l}|\mathbf{b}=i)} + \sum_{i=1}^{B} \mathbf{w}_i \mathbb{E}_{\mathbf{l} \sim q(\mathbf{l}|\mathbf{b}=i)} \log \mathbf{w}_i \\ &= \sum_{i=1}^{B} \mathbf{w}_i \cdot \text{KL}\left( q(\mathbf{l}|\mathbf{b}=i) \parallel \sum_{i=1}^{B} \mathbf{w}_i q(\mathbf{l}|\mathbf{b}=i) \right) + \sum_{i=1}^{B} \mathbf{w}_i \log \mathbf{w}_i \\ &\geq \sum_{i=1}^{B} \mathbf{w}_i \log \mathbf{w}_i. \end{aligned} \tag{21}$$

Thus, the minimization of Eq. (21) is equivalent to minimizing a generalized form of Jensen–Shannon divergence among cell-embedding distributions of multiple batches, the global minimum being $\sum_{i=1}^{B} \mathbf{w}_i \log \mathbf{w}_i$, reached if and only if $q(\mathbf{l}|\mathbf{b}=i) = q(\mathbf{l}|\mathbf{b}=j), \forall i, j$. In that case, the batch discriminator can no longer tell which batch a cell belongs to based on its embedding $\mathbf{l}$, and the only best guess it can give is the overall proportion of each batch.

Note that in practice, model training balances between $\mathcal{L}_{\text{base}}$ and pure batch alignment described above. Aligning cells of the same type induces minimal or no cost in $\mathcal{L}_{\text{base}}$, while improperly aligning cells of different types could cause $\mathcal{L}_{\text{base}}$ to rise dramatically since the model cannot generate drastically different gene expression profiles from the same $\mathbf{l}$.

The performance improvement delivered by this adversarial strategy in the batch-effect correction benchmark suggests that the gradient from both batch discriminators and decoder may provide better guidance to align different batches than hand-crafted alignment strategies like CCA[32] and MNN[6].

If multiple levels of batch effect exist, e.g., within-dataset and cross-dataset, we use an independent batch discriminator for each component, providing extra flexibility.

**Gene selection**. Most informative genes for individual datasets were selected using the Seurat[32] function "FindVariableGenes" (v2.3.3). Briefly, it plots the variance–mean–ratio of each gene against its mean expression level, group genes into several bins based on their mean expression levels, and then select genes with significantly higher variance in each bin. We tested a wide range of cutoff values and found that the Cell BLAST performance is relatively stable as the number of selected genes varies from 500–5000 (Supplementary Fig. 1d). For all other experiments, we set the argument "binning.method" to "equal_frequency", and left other arguments as default. If within-dataset batch effect exists, genes are selected independently for each batch and then pooled together. By default, a gene is retained if it is selected in at least 50% of batches. In most cases, 500–1000 genes are selected. For scmap and CellFishing.jl, which provide their own gene selection methods, we used these recommended gene selection methods.

When merging multiple datasets (e.g., in batch correction evaluation), we first merged expression matrices based on gene names. If datasets to be merged are from different species, Ensembl ortholog[49] information was used to map genes to ortholog groups before merging. Then, to obtain informative genes in merged datasets, we took the union of informative genes from each dataset, and then intersected the union with the intersection of detected genes from each dataset.

**Benchmarking dimension reduction**. PCA, tSNE[8], and ZINB-WaVE[37] were performed using R packages irlba[50] (v2.3.3), Rtsne (v0.15), and zinbwave (v1.6.0), respectively. UMAP[51] and scPhere[35] were performed using Python package umap-learn (v0.3.8) and scPhere (v0.1.0), respectively. ZIFA[52] (v0.1), Dhaka[26] (v0.1), DCA[7] (v0.2.2), scVI[11] (v0.2.3), scScope[53] (v0.1.5), and SAUCIE[54] source code were downloaded from their Github repositories. For ZIFA and scScope, we removed hard-coded random seeds and added options for manually setting them. For scVI, minor changes were made to address PyTorch GPU compatibility issues. All the modified packages are available in our Github repository (https://github.com/gao-lab/Cell_BLAST/tree/master/local).

For PCA, ZIFA, Dhaka, scScope, and SAUCIE, data were logarithm transformed after normalization and adding a pseudocount of 1. tSNE and UMAP were applied on the first 50 principal components of PCA. ZINB-WaVE, DCA, scVI, scPhere, and Cell BLAST were fitted on raw data. Hyperparameters of all methods above were left as default. The input gene sets for all methods are identical.

Cell-type nearest-neighbor mean average precision (MAP) was computed with $K$-nearest neighbors of each cell based on low-dimensional space Euclidean distance. If we denote the cell type of a cell as $y$, and the cell types of its ordered nearest neighbors as $y_1, y_2, \ldots y_K$, the average precision (AP) for that cell can be computed as:

$$\text{AP} = \begin{cases} \dfrac{\sum_{k=1}^{K} 1_{y=y_k} \frac{\sum_{k=1}^{k} 1_{y=y_k}}{k}}{\sum_{k=1}^{K} 1_{y=y_k}}, & \text{if } \sum_{k=1}^{K} 1_{y=y_k} > 0 \\ 0, & \text{otherwise} \end{cases} \tag{22}$$

where $1_y = y_k$ is an indicator function that evaluates to 1 when $y = y_k$ and 0 when

$y \neq y_k$. Denote the average precision for the $i$th cell as $\text{AP}^{(i)}$, mean average precision is then given by:

$$\text{MAP} = \frac{1}{N}\sum_{i=1}^{N} \text{AP}^{(i)}. \qquad (23)$$

Note that when $K = 1$, MAP reduces to nearest-neighbor accuracy. We set $K$ to 1% of the total cell number throughout all benchmarks. The range of MAP is 0–1, with higher values indicating better cell-type resolution.

We tested each method on a wide range of dimensionalities (except for tSNE, UMAP, and SAUCIE, which are limited to dimensionality = 2), and picked the optimal dimensionality for each method based on its average performance across all datasets. All experiments were repeated 16 times with different random seeds.

**Benchmarking batch-effect correction.** ComBat[55], MNN[6], CCA[32], CCA anchor[20], Harmony[56], scPhere[35], and scVI[11] were performed using R packages sva[57] (v3.32.1), scran[58] (v1.6.9), Seurat (v2.3.3), Seurat (v3.0.2), harmony (v1.0), and Python package scPhere (v0.1.0), scVI (v0.2.3), respectively. SAUCIE[54] was performed using the "SAUCIE.py" script provided in their Github repository. Hard-coded random seeds in Seurat (both v2.3.3 and v3.0.2) were removed to reveal actual stability. The modified versions of Seurat are available in our Github repository. ComBat, MNN, CCA, CCA anchor, and SAUCIE were all performed on normalized and logarithm transformed data, while scVI, scPhere, and Cell BLAST were fitted on raw data. For MNN alignment, we set the argument "cos.norm.out" to false. Harmony was performed on PCA of the normalized and logarithm transformed data. All other hyperparameters were left as default. The input gene sets for all methods are identical.

We use Seurat alignment score[32] to evaluate the extent of batch mixing, and use mean average precision to quantify the preservation of biological signal. Mean average precision was computed as described in the "Benchmarking dimension reduction" section. Seurat alignment score was computed as described in the original CCA paper[32]:

$$\text{Seurat alignment score} = 1 - \frac{\bar{x} - \frac{k}{N}}{k - \frac{k}{N}}, \qquad (24)$$

where $\bar{x}$ is the average number of cells belonging to the same batch in $k$-nearest neighbors (with different batches subsampled to the same size), and $N$ is the number of batches. The range of Seurat alignment score is 0−1, with higher scores indicating better dataset mixing.

For best comparability, mean average precision and Seurat alignment score should be computed in low-dimensional space. For CCA, Harmony, SAUCIE, scPhere, scVI, and Cell BLAST, corrected data are already in low-dimensional spaces, while for ComBat, MNN, and CCA anchor which return corrected expression matrices, PCA was applied to reduce the dimensionality afterwards.

For Cell BLAST, we tested different values of $\lambda_b$, which determines batch alignment strength, and fixed embedding dimensionality at 10 (as determined in the previous dimensionality reduction benchmark). Other methods do not provide a similar hyperparameter that specifically controls batch alignment strength, so we tested on a wide range of dimensionalities (either the dimensionality of the method itself, or the post hoc PCA dimensionality). Optimal configuration for each method was picked based on the average of sum of mean average precision and Seurat alignment score across all datasets. We noted that the range of mean average precision among different methods is much smaller than that of Seurat alignment score, so we assigned larger weight (0.9) to mean average precision and smaller weight (0.1) to Seurat alignment score to balance them. All benchmarked methods were repeated 16 times with different random seeds.

**Posterior visualization and comparison.** We fitted scVI[11] and Cell BLAST on the "Baron_human"[12], "Adam"[13], and "Guo"[59] dataset with identical gene sets and two-dimensional embedding space, and then sampled from their variational posterior and true posterior distributions, respectively. For Cell BLAST, the variational posterior samples were obtained as described in the "Approximate posterior and model optimization" section. To sample from the scVI variational posterior, we took the posterior mean ($\mu$) and variance ($\sigma^2$) from the encoder output, and then sampled from $N(\mu,\text{diag}(\sigma^2))$. To sample from the true posteriors of scVI and Cell BLAST, we used the Metropolis Hastings Markov chain Monte Carlo (MCMC) algorithm[60].

To examine the posterior distributions of cells along the border of the cell types of interest, we fitted Gaussian kernel support vector machines (SVM) to classify the cell types in the two-dimensional embedding space, and then used the support vectors as border cells. Support vectors with small decision-function values were excluded because their cell-type annotation can be unreliable. To summarize the shape of posterior distribution in this region, we aggregated posterior samples from multiple cells after subtracting the posterior point estimate to center them. SVD was then performed on the aggregated samples to reveal the major and minor axes of variation.

**Distance metric ROC analysis.** Distance metric ROC analysis was performed using the pancreatic group in Supplementary Table 3. We fitted scVI[11] and Cell BLAST models on the reference datasets using the same gene set (with batch-effect correction), and then applied the fitted models to all datasets including positive and negative queries to obtain samples from the posterior distributions, as well as cell embeddings (posterior point estimates). We then subsampled query datasets to equal sizes, and searched for 50 nearest reference cells for each query cell. A query-reference nearest-neighbor pair is defined as positive if the query cell and reference cell are of the same cell type, and negative otherwise. Each distance metric was then computed on the nearest-neighbor pairs and used as predictors for positive/negative pairs. AUC values were computed for each distance metric based on the ROC curves. Both scVI and Cell BLAST were repeated 16 times with different random seeds.

**Posterior distance estimation.** We evaluated cell-to-cell similarity based on a custom distance distance between their posterior distributions in the embedding space, which we termed normalized projection distance (NPD). To obtain a robust estimation of the distribution distance with a small number of posterior samples, we project the posterior samples of two cells onto the line connecting their posterior point estimates, converting the multi-dimensional posterior distributions to scalar distributions. NPD is then computed based on Wasserstein distance of normalized projection values:

$$\text{NPD}(p, q) = \frac{1}{2}\left(W_1\left(\mathbb{Z}_p(p), \mathbb{Z}_p(q)\right) + W_1\left(\mathbb{Z}_q(p), \mathbb{Z}_q(q)\right)\right); \qquad (25)$$

where $p$ and $q$ are scalar projection values, and

$$W_1(u, v) = \inf_{\pi \in \Gamma(u,v)} \int |x - y| d\pi(x, y); \qquad (26)$$

$$\mathbb{Z}_u(v) = \frac{v - \mathbb{E}(u)}{\sqrt{\text{var}(u)}}. \qquad (27)$$

Normalization effectively rescales the local embedding space so that the cell-to-cell distances reflect true biological similarity more accurately (see Supplementary Fig. 8a for an intuitive illustration). By default, 50 samples from the posterior are used to compute NPD, which produces sufficiently accurate results (Supplementary Fig. 8b).

**Cell querying using posterior distance.** We first apply the pretrained models on query cells to obtain cell embeddings (posterior point estimates) and samples from posterior distributions. The definition of NPD does not imply an efficient nearest-neighbor searching algorithm. To increase speed, we first perform nearest-neighbor search based on Euclidean distance between cell embeddings to obtain the initial query hits, which is highly efficient in the low-dimensional embedding space. We then compute posterior distances only for these nearest neighbors. The empirical distribution of posterior NPD for a dataset is obtained by computing posterior NPD on randomly selected pairs of cells in the reference dataset. Empirical $P$-values of query hits are computed by comparing the posterior NPD of query hits to this empirical distribution.

We note that even with the strategy described above, querying with single models still occasionally leads to many false-positive hits when cell types on which the model has not been trained (non-existent cell types in the reference) are provided as query. This is because embeddings of such untrained cell types are more or less random, and could localize close to reference cells by chance. We reason that the embedding randomness of untrained cell types could be utilized to identify and correctly reject them. In practice, we train multiple models with different starting points (as determined by random seeds) and compute query hit significance for each model. A query hit is considered significant only if it is consistently significant across multiple models. We found that the strategy improves specificity significantly without sacrificing sensitivity (Supplementary Fig. 8d).

**Hit-based predictions.** Predictions can be made for query cells using the existing annotations of significant hit cells. A minimal number (by default 2) of significant hit cells ($P$-value smaller than a specified cutoff, by default 0.05) are required for prediction to be made, otherwise the query cell is rejected. For discrete annotations, e.g., cell type, majority voting among the significant hit cells is used. We require that the majority votes exceed a threshold (by default 50% of all significant hits) for confident prediction to be made, otherwise the query cell is labeled as "ambiguous". For continuous annotations like the cell fate probabilities, we use the arithmetic average of hit cell annotations.

**Online tuning.** When significant batch effect exists between reference and query, we support further aligning query data with the reference data in an online-learning manner. All components in the pretrained model, including the encoder, decoder, prior discriminators, and batch discriminators, are retained. The reference-query bias is added as an extra batch effect to be corrected using adversarial batch alignment. Specifically, a new discriminator dedicated to the reference-query batch effect is added, and the decoder is expanded to accept an extra one-hot indicator for reference and query. The expanded model is then fine-tuned using the combination of reference and query data. Two precautions are taken to prevent a decrease in specificity caused by over-alignment. First,

adversarial alignment loss is constrained to cells that have mutual nearest neighbors[6] between reference and query data in each SGD minibatch. Second, we penalize the deviation of tuned model weights from the original weights. Among all experiments in the paper, online tuning is only used when performing cross-species querying between the "Velten" and "Tusi" dataset in the hematopoietic progenitor study.

**Benchmarking query-based cell typing**. Cell ontology annotations in ACA were used as ground truth. Cells without cell ontology annotations were excluded in the analysis. For clarity, we use the notation $\mathcal{Q}$ for the complete set of query cells, $\mathcal{T}_+$ for the set of positive cell types (cell types existent in the reference), $\mathcal{T}_-$ for the set of negative cell types (cell types not existent in the reference), $f$ and $g$ as functions that map cells to their actual cell types and predicted cell types, respectively.

For evaluating cell-type prediction accuracy, we manually compiled expected prediction matrices $\mathbf{M}$ based on descriptions in the original publications as well as cell ontology (Supplementary Data 3). This improves the estimation of prediction accuracy by accommodating negative queries (with "reject" as correct prediction) as well as reasonable predictions that would be otherwise considered incorrect, e.g., "blood vessel endothelial cell" being predicted as "lung endothelial cell" when using pulmonary references, "cytotoxic T cell" being predicted as "T cell" when using pancreatic references (immune cells annotated in lower resolution in the pancreatic studies because they were not the research focus), etc. Rows of $\mathbf{M}$ represent actual cell types, and columns represent predictions. A cell-type prediction is considered correct if the corresponding entry in $\mathbf{M}$ is 1, and incorrect otherwise. Prediction accuracy for each cell type $t$ is computed as:

$$\text{Accuracy}(t) = \frac{\left|\left\{q | q \in \mathcal{Q}, f(q) = t, \mathbf{M}_{f(q),g(q)} = 1\right\}\right|}{\left|\{q | q \in \mathcal{Q}, f(q) = t\}\right|}. \tag{28}$$

Mean balanced accuracy (MBA) can then be computed as the equally weighed arithmetic mean of prediction accuracy of each cell type:

$$\text{MBA}(\mathcal{T}) = \frac{1}{|\mathcal{T}|} \sum_{t \in \mathcal{T}} \text{Accuracy}(t), \tag{29}$$

where $\mathcal{T}$ denotes the set of cell types under consideration. In our evaluation (Fig. 2b, c; Supplementary Fig. 9a, c), we reported both $\text{MBA}(\mathcal{T}_+)$ and $\text{MBA}(\mathcal{T}_-)$. For the average MBA, we used $\frac{\text{MBA}(\mathcal{T}_+)+\text{MBA}(\mathcal{T}_-)}{2}$ rather than $\text{MBA}(\mathcal{T}_+ \cup \mathcal{T}_-)$, because of the imbalance nature of query cell types (i.e., much larger number of negative cell types than positive cell types would dominate the latter).

Scmap[2] and CellFishing.jl[4] were performed using R package scmap (v1.6.0) and Julia package CellFishing (v0.3.0), respectively. For all experiments with scmap, we used the scmap-cell algorithm. For each method, its own recommended gene selection method was used. Default cutoff of scmap (cosine similarity = 0.5) and Cell BLAST ($P$-value = 0.05) were used to reject unmatched cells. As CellFishing.jl does not come with a query-based prediction method, we used the same strategy as Cell BLAST, with Hamming distance = 120 as cutoff determined from grid searching for best balance between correctly predicting positive types, i.e., $\text{MBA}(\mathcal{T}_+)$, and rejecting negative types, i.e., $\text{MBA}(\mathcal{T}_-)$ across all four dataset groups (Supplementary Fig. 9c). Of interest, the grid searching found that scmap shows slightly higher average MBA at a stricter cutoff (cosine similarity = 0.6) than its default one (0.5), but still lower than that of Cell BLAST (Supplementary Fig. 9c).

All benchmarked methods were repeated four times with different random seeds. Several other cell querying tools (CellAtlasSearch[3], scQuery[28], scMCA[61]) were not included in our benchmark because they do not support custom reference datasets.

**Benchmarking querying speed**. To evaluate the scalability of querying methods, we constructed reference datasets of varying sizes by subsampling from the 1 M mouse brain dataset[16]. For query data, the "Marques" dataset[62] was used. The input gene sets for all methods are identical. Benchmarking was performed on a workstation with 40 CPU cores, 100GB RAM and GeForce GTX 1080Ti GPU. For all methods, only the querying time was recorded, not including the time consumed to index the references.

**Identifying cell typing-important genes**. For each dataset, we used each cell type in turn as query data, and the remaining cells as reference data to perform Cell BLAST querying. The embedding space deviation of query cells from their reference hits ($\Delta \mathbf{l} \in \mathbb{R}^D$) are propagated back to the original gene space via the gradient of the encoder neural network:

$$\Delta \tilde{\mathbf{x}} = \left(\frac{\partial \mathbf{l}}{\partial \tilde{\mathbf{x}}}\right)^\top \cdot \Delta \mathbf{l}, \tag{30}$$

where $\frac{\partial \mathbf{l}}{\partial \tilde{\mathbf{x}}} \in \mathbb{R}^{D \times |\mathcal{G}^*|}$, $\Delta \tilde{\mathbf{x}} \in \mathbb{R}^{|\mathcal{G}^*|}$. Deviation $\Delta \mathbf{l}$ is first normalized to unit vectors.

By averaging $\Delta \tilde{\mathbf{x}}$ across all reference hits of all query cells, we obtain the genewise gradient from surrounding cell types towards the cell type of interest. We also average $\Delta \tilde{\mathbf{x}}$ across multiple Cell BLAST models to improve reliability. Large gradient value for the $i$th gene ($\Delta \tilde{\mathbf{x}}_i$) indicates that higher expression of the gene

causes cell embedding to move towards the cell type of interest. In other words, the gene is an important feature for the specific cell type. To test whether gene importance as derived above is consistent with prior knowledge, we downloaded the manually curated cell-type marker list from PanglaoDB[15] (Jan 21, 2020). Gene rankings based on the gradient values were fed to the GSEA prerank algorithm[63] as implemented in gseapy (v0.9.16) to test whether PanglaoDB markers are significantly enriched in genes with larger gradient values. PanglaoDB markers were first filtered based on whether mean expression in the asserted cell type is among the top three in the reference dataset. FDR correction was applied to control false positives in multiple tests. Only cell types in which the number of PanglaoDB marker genes intersecting $\mathcal{G}^*$ is greater than 10 were tested. We were able to perform the above analysis for three of the four reference datasets in the cell-querying benchmark (Supplementary Table 3). The mammary dataset was discarded because only one cell type exists in the PanglaoDB marker list.

**Application to trachea datasets**. We first removed cells labeled as "ionocytes" in the "Montoro_10x"[17] dataset and used Seurat function "FindVariableGenes" to select informative genes using the remaining cells.

For Cell BLAST, four models with different starting points were trained on the tampered "Montoro_10x" dataset. We used the default cutoff of empirical $P$-value > 0.05 to reject query cells from the "Plasschaert"[18] dataset. We clustered the rejected cells using Louvain community detection[64] on shared nearest-neighbor (SNN) graph constructed based on cell embeddings.

For scmap[2], we used the default cutoff of cosine similarity < 0.5 to reject query cells. Louvain community detection on SNN graph was also used to cluster rejected cells, but we used cosine similarity to construct the SNN graph instead. For other cell-querying methods, we used their overall optimal cutoff as determined in the query-based cell typing benchmark.

Ionocyte enrichment ratio in Supplementary Fig. 14g was computed as follows:

$$\text{Enrichment ratio} = \frac{\#\text{Ionocytes}/\#\text{All cells}}{\#\text{Rejected ionocytes}/\#\text{Rejected cells}}. \tag{31}$$

Genes in the expression heatmaps of Supplementary Figs. 11, 14e, 15a were selected in two ways. For markers of the retained cell types (cell types except ionocyte), we directly obtained known marker genes from the "Montoro" study. For markers of the rejected cell clusters (only in Supplementary Figs. 11, 14e), we performed one-versus-rest differential expression analysis using two-sided Wilcoxon rank-sum test to find genes highly expressed in each cluster, and selected the most significant genes with Bonferroni corrected $P$-value < 0.05. GO enrichment analysis was conducted using Metascape[65].

**Application to hematopoietic progenitor datasets**. For within- "Tusi"[19] querying, we trained four models using only cells from sequencing run 2, while cells from sequencing run 1 were used as query cells. Population balance analysis (PBA) inferred cell fate probabilities provided by the authors were used as the ground truth. Below, we denote the cell fate probabilities as a seven-dimensional vector $\mathbf{p} \in \mathbb{R}^7$. Each dimension $0 \leq \mathbf{p}_i \leq 1$, $i = 1, 2, \ldots, 7$ corresponds to the probability that the cell will differentiate into a specific terminal cell lineage. The seven probabilities are normalized and form a probability distribution, i.e., $\sum_{i=1}^{7} \mathbf{p}_i = 1$.

The visualization in Fig. 3d–e, Supplementary Fig. 16c–g are obtained by applying SPRING[66] to the cell embeddings. Each cell lineage is assigned a distinct color. Each cell is then colored according to the cell lineage of largest differentiation probability. In addition, the saturation of each cell is determined by the cell fate entropy $\mathbb{H}[\mathbf{p}] = \sum_{i=1}^{7} -\mathbf{p}_i \log \mathbf{p}_i$. Thus, cells committed to a specific lineage would have more saturated colors, while undifferentiated cells would appear gray.

For Cell BLAST, the cell fate predictions are obtained as described in the "Predictions based on query hits" section. Scmap[2] and CellFishing.jl[4] do not support predicting continuous variables, so we extended them using the same strategy as Cell BLAST. CCA anchor[20] natively supports transferring continuous variables in its "TransferData" function. scANVI[21] was designed as a semi-supervised classifier that returns the probability distribution over different classes. We used reference cells with maximal lineage probability > 0.5 as labeled data, and the remaining cells as unlabeled data when train scANVI. The batch effect between reference and query data was also specified to be corrected. The resulting class distribution in the query data was then interpreted as transferred lineage probability.

Jensen–Shannon divergence (JSD) between true and predicted cell fate distributions was computed as below:

$$\text{JSD}(\mathbf{p} \parallel \mathbf{q}) = \frac{1}{2} \cdot \sum_{i=1}^{7} \mathbf{p}_i \log \frac{\mathbf{p}_i}{\frac{\mathbf{p}_i + \mathbf{q}_i}{2}} + \mathbf{q}_i \log \frac{\mathbf{q}_i}{\frac{\mathbf{p}_i + \mathbf{q}_i}{2}}, \tag{32}$$

where $\mathbf{p}$ and $\mathbf{q}$ denote the ground truth and predicted cell fate distribution, respectively.

For cross-species querying between "Tusi" and "Velten"[22], we mapped human genes to mouse orthologs before merging. The online-tuning mode of Cell BLAST was used to correct for the cross-species batch effect. Due to the significant cross-species batch effect, CellFishing.jl produced excessive number of false negative rejections under the previously determined cutoff of Hamming distance = 120, so

we used a more permissive cutoff of Hamming distance = 170 to achieve a smaller rejection rate comparable to scmap and Cell BLAST. Since no ground truth cell fate probabilities are available for the "Velten" dataset, we used the Spearman correlation between predicted cell fate probabilities and expression level of known cell lineage markers from the "Velten" study (Supplementary Fig. 17) as a quantitative metric. We report the average of three marker genes of the highest correlation coefficient for each cell lineage in Fig. 3f. Since the "Velten" dataset does not distinguish between the monocyte (M) and the dendritic cell (D) lineage, we merged them into a single M/D lineage by summing up the PBA probabilities.

**ACA database construction.** We searched Gene Expression Omnibus (GEO)[67] using the following search term:

```
(
 expression profiling by high throughput
sequencing[DataSet Type] OR
 expression profiling by high throughput
sequencing[Filter] OR
 high throughput sequencing[Platform
Technology Type]) AND
 gse[Entry Type] AND
 (
 single cell[Title] OR
 single-cell[Title]) AND
 (2013[Publication Date]: 3000[Publication
Date]) AND
 supplementary[Filter]
```

Datasets in the Hemberg collection (https://hemberg-lab.github.io/scRNA.seq.datasets/) were merged into this list. Only animal single-cell transcriptomic datasets profiling samples of normal conditions were selected. We also manually filtered small-scale or low-quality data. In addition, several other high-quality datasets missing in the previous list were included for comprehensiveness.

The expression matrices and metadata of selected datasets were retrieved from GEO, supplementary files of the publication or by directly contacting the authors. Metadata were further manually curated by adding additional descriptions in the paper to acquire the most detailed information of each cell. We unified raw cell-type annotation by Cell Ontology[68], a structured controlled vocabulary for cell types. Closest Cell Ontology terms were manually assigned based on the Cell Ontology description and context of the study.

**Building reference panels for the ACA database.** Two types of searchable reference panels are built for the ACA database. The first consists of individual datasets with dedicated models trained on each, while the second consists of datasets grouped by organ and species, with models trained to align multiple datasets profiling the same species and same organ.

Data preprocessing follows the same procedure as in previous benchmarks. Both cross-dataset batch effect and within-dataset batch effect are manually examined and corrected when necessary. For the first type of reference panels, datasets too small (typically <1000 cells sequenced) are excluded because of insufficient training data. These datasets are still included in the second type of panels, where they are trained jointly with other datasets profiling the same organ in the same species. For each reference panel, four models with different starting points are trained.

**BLAST2CO.** BLAST2CO can perform Cell Ontology (CL) inference for query cells as long as the reference datasets are annotated with Cell Ontology (see Supplementary Fig. 19a for a schematic diagram of the BLAST2CO workflow). For each CL existent in the hit cells, BLAST2CO calculates its confidence score by summing up the similarity (defined as $1 - "P\text{-value}"$) of all hit cells belonging to the particular CL, normalized by the sum in all hits. The confidence scores are assigned to corresponding nodes in the CL graph, and then propagated to parent nodes along the graph. CLs with confidence scores exceeding a given threshold (by default 0.5) are retained to form a subgraph. Finally, the leaf node of the subgraph with maximal confidence score is chosen as final prediction. We additionally require the longest path length to root of the predicted CLs to exceed a given threshold, to guarantee that the final predictions are not too coarse. If more than one CLs match the above conditions, the query cell is labeled as "ambiguous". If no CL matches the above conditions, the query cell is labeled as "rejected".

In the human kidney case study (Supplementary Fig. 19b–e), both the incorporation of hit similarity and ontology structure contribute to the improved predictions. On the one hand, hits for many "stromal cells" consisted of the same number of "fibroblasts" and "mesangial cells", causing majority voting to fail. BLAST2CO was able to break the tie by considering the similarity of each hit. On the other hand, hits for many "epithelial cells of nephron" and some "stromal cells" consisted of several different yet related CLs. By leveraging prior knowledge encoded in the CL graph, BLAST2CO identified the most confident ancestor CLs for these cells ("epithelial cell" for "epithelial cell of nephron", "connective tissue cell" for "stromal cell").

To quantitatively evaluate the performance of Cell Ontology inference, we designed an ontology-aware accuracy metric called CL accuracy. Denote CL terms in the CL directed acyclic graph as $c_i \in C$, where $C$ is the set of all CL terms. Let $\prec$ be the "is_a" relationship, i.e., $c_i \prec c_j$ denotes that fact that $c_i$ is a subtype of $c_j$. Note that $\prec$ includes indirect relationships, i.e., $c_i \prec c_j$, $c_j \prec c_k$ leads to $c_i \prec c_k$. Let $c_i \preccurlyeq c_j$ denote $c_i \prec c_j$ or $c_i = c_j$. For simplicity, let $c_i \preccurlyeq C^*$ denote the fact that $\exists c_j \in C^*, c_i \preccurlyeq c_j$, where $C^* \subseteq C$ is an arbitrary subset of CL terms. Similarly, $C^* \preccurlyeq c_i$ denotes $\exists c_j \in C^*, c_j \preccurlyeq c_i$. Further, we use $\nabla_{c_i} = \left\{ c_j | c_j \preccurlyeq c_i \right\}$ to denote the descendent set of $c_i$.

Let $C_{\text{ref}}$ and $C_{\text{query}}$ be the set of all CL terms existent in the reference and query dataset, respectively. When computing CL accuracy, we first determine whether query CLs are positive or negative cell types. A query CL $c_i \in C_{\text{query}}$ is considered positive if $c_i \preccurlyeq C_{\text{ref}}$ or $C_{\text{ref}} \preccurlyeq c_i$. Otherwise the query CL is considered negative. When a positive query CL $c_i$ is predicted as $c_j$ (note that in BLAST2CO, it is possible that $c_j \notin C_{\text{ref}}$), CL accuracy is computed as:

$$\text{CL accuracy}\left(c_i, c_j\right) = \begin{cases} \frac{1}{|C_I|} \sum_{c_k \in C_I} \frac{\left| \nabla_{c_k} \right|}{\left| \nabla_{c_j} \right|}, & c_i \prec c_j \\ 1, & c_j \preccurlyeq c_i \\ 0, & \text{otherwise} \end{cases} \quad (33)$$

where $C_I$ is the set of intermediate CL terms between $c_i$ and $c_j$ which are also theoretically inferable from $C_{\text{ref}}$:

$$C_I = \left\{ c_k | c_i \preccurlyeq c_k \preccurlyeq c_j, C_{\text{ref}} \preccurlyeq c_k \right\}. \quad (34)$$

For negative query CLs, CL accuracy = 1 if the cells are rejected, otherwise CL accuracy = 0. CL MBA can then be calculated by averaging the above CL accuracy across different CL terms.

**Web interface.** For conveniently performing and visualizing Cell BLAST analysis, we built a one-stop Web interface. The client-side was made from Vue.js, a single-page application Javascript framework, and D3.js for cell ontology visualization. We used Koa2, a web framework for Node.js, as the server side. The Cell BLAST Web portal with all accessible curated datasets is deployed on Huawei Cloud.

**Reporting summary.** Further information on research design is available in the Nature Research Reporting Summary linked to this article.

## Data availability
All scRNA-seq datasets used in this study were obtained from public data repositories, with detailed information including accession codes and URLs available in Supplementary Data 2. Source data for the benchmark experiments are available in Supplementary Data 4. Curated datasets in ACA are available through our Web portal https://cblast.gao-lab.org/download.

## Code availability
The full package of Cell BLAST is available at https://cblast.gao-lab.org. Code necessary to reproduce results in the paper is deposited at https://github.com/gao-lab/Cell_BLAST. To ensure reproducibility, all benchmarks and tests are assembled using Snakemake, and environment configuration files are provided. The Cell BLAST python package is also available as Supplementary Software.

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

## Acknowledgements

The authors thank Drs. Zemin Zhang, Cheng Li, Letian Tao, Jian Lu and Liping Wei at Peking University for their helpful comments and suggestions during the study. This work was supported by funds from the National Key Research and Development Program (2016YFC0901603), the China 863 Program (2015AA020108), as well as the State Key Laboratory of Protein and Plant Gene Research and the Beijing Advanced Innovation Center for Genomics (ICG) at Peking University. The research of G.G. was supported in part by the National Program for Support of Top-notch Young Professionals. Part of the analysis was performed on the Computing Platform of the Center for Life Sciences of Peking University and supported by the High-performance Computing Platform of Peking University.

## Author contributions

G.G. conceived the study and supervised the research; Z.J.C. and L.W. contributed to the computational framework and data curation; S. L., Z.J.C., and D.C.Y designed, implemented and deployed the website; Z.J.C. and G.G. wrote the paper with comments and inputs from all coauthors.

## Competing interests

The authors declare no competing interests.
