## [Peer Review File · Nature Communications]

Reviewers' Comments:

Reviewer #1:

Remarks to the Author:

The paper presents cell BLAST, a deep generative model to automatically identify cell types by searching a reference database.

As more cells are sequenced and made available to the public, Cell BLAST (and similar tools) could be useful for us to rapidly interpret scRNA-seq results.

The paper is well written and easy to follow.

I have some major concerns with the method and the results, which need to be addressed.

a) Since 2017, deep generative models have been used and established as valuable for different scRNA-seq data analysis tasks, e.g., dimension reductions, batch correction, and cell type annotations. In this manuscript, the authors use Adversarial Autoencoders (AAE), which is more flexible than variational autoencoders (VAE) but has more parameters and harder to train. In this manuscript, I don't see the advantages of using AAE to address any specific shortcomings of VAE in scRNA-seq data analysis. In several benchmark tests, e.g., dimension reductions, scVI (based on VAE) using a 5D latent space performs similarly to Cell Blast using 10D latent space (Supp. Fig.2).

b) I appreciate the authors put their tool online for people to use. A recent paper (PMID: 31500660) systematically compared 22 algorithms and found off-the-shelf SVM classifier works very well in practice. Cell BLAST did not perform so well compared to several classifiers such as SVM, RF, and LDA. Moreover, even scVI performed better than Cell BLAST in general. Finally, Cell BLAST is time-consuming to run. From this comparison paper, I don't see much of the values of using Cell Blast in practice. I do think that the authors should comment on and ideally repeat the Cell B last experiments in the comparison paper.

c) From the experiments presented in this manuscript, it seems that Cell BLAST works well, and the p-values provided by Cell BLAST is useful for users to interpret the results. However, as deep learning models are 'black box' methods and are prone to adversarial attacks, extra information is helpful. For example, the authors can compute the ROCs of marker genes in discriminating a cell type from other cells.

d) The authors introduced an adversarial batch correction term in the objection function and a weight lambda parameter to balance its contribution to the objective function. This parameter could be data-dependent. In Fig.1, it seems that Cell Blast did a decent job in batch-correction for the specific dataset. However, we still see strong batch effects in other datasets (e.g., Supp. Fig. 3g, Supp. Fig. 4a). Therefore, to prove the robustness and effectiveness of this adversarial batch-correction strategy, the authors need to do more side-by-side comparisons (e.g., with scVI) across different datasets.

e) Currently, people have collected large numbers of scRNA-seq datasets, e.g., the panglaodb (<https://panglaodb.se/index.html>) has about 5.6 million collected and annotated cells. Moreover, the authors provide the marker genes for cell annotations. I think these maker genes could be useful for interpreting Cell Blast results (as in c).

f) The authors used a 'normalized projection distance' for cell query and shows it improves Cell Blast query but has almost no effects on scVI results. It's not clear to me why this is the case. Have the authors do the comparisons on different datasets? How the weight (beta) of the KL divergence term in the VAE objective function influence the results?

In summary, I think that Cell BLAST is an interesting and useful tool, especially for people with

limited computational skills. However, I'm not convinced that it significantly improves similar tools for automatic cell annotations and for robust batch corrections given extra parameters and the results presented in this paper and in the comparison paper(PMID: 31500660).

Reviewer #2:

Remarks to the Author:

The manuscript by Cao et al. presents Cell Blast, a method and webtool to automatically assign cell identities by co-embedding single cell RNA-seq data with annotated reference data. The authors propose a generative model based on neural-networks to learn non-linear embeddings of high dimensional single cell RNA-seq data. The authors evaluated their method in terms of the accuracy of the resulting low dimensional embeddings, batch effect correction, and cell identification.

The proposed method is interesting and, in particular, the webtool and the curated reference data can be a valuable resource for the community. However, the manuscript would benefit from addressing a number of issues in greater detail - see below.

Major comments

- The paper can greatly benefit from a figure illustrating the structure of the generative model used by the authors to improve readability of the methods section.
- The model presented in this paper is similar to scVI. The major difference appears to be the reliance on adversarial training between the encoder and the discriminator networks while scVI relies on variational inference. In that sense, a proper discussion of the similarities and differences between the proposed model and scVI is important.
- I disagree with the authors regarding the ability of scVI to perform cell querying (Supplementary Table 1). A modified version of scVI (scANVI) allows annotation of cells based on the posterior distribution, much like Cell Blast. This is described in a preprint: <https://www.biorxiv.org/content/10.1101/532895v1> and on GitHub: <https://nbviewer.jupyter.org/github/YosefLab/scVI/blob/master/tests/notebooks/annotation.ipynb>. As such, I believe a proper comparison of both methods in terms of cell querying is important.
- One of the strong points of the paper is the extensive curated collection of reference data. However, the rapid growth of scRNA-seq data will deem any reference outdated very quickly. What are the authors thoughts regarding maintaining their curated reference atlas? And more importantly, how does updating the atlas affect the pretrained model?
- I got confused in the explanation of the stochastic autoencoder part (equations 6 – 9). In particular, how is it possible to skip the Poisson sampling. Please explain.
- The authors rely on "Cell type nearest neighbor mean average precision (MAP)" to evaluate the accuracy of dimensionality reduction. This measure only evaluates how homogenous are the resulting low dimensional embeddings, with no regard to the correspondence of the neighborhood structure between the high and low dimensional spaces. My concern here stems from the fact that most of the cell labels are derived from manual annotations based on overlaying marker gene expression on low dimensional embeddings. Perhaps the authors should consider a more appropriate measure, such as: <https://www.biorxiv.org/content/10.1101/786269v1>
- I am curious why did the authors decide to exclude tSNE and UMAP from the comparison of dimensionality reduction methods, while both are widely used within the single cell community.
- While comparing the batch effect correction, the authors used PCA to obtain low dimensional embeddings for the corrected data of ComBat, MNN, and CCA anchor. This seems a bit unfair given that PCA on its own did not perform well, based on Figure 1b. Why not calculate MAP for these methods on the high dimensional data directly?
- When comparing the performance of cell querying (Figure 1c and 1d), the authors report only the TP and TN values. It is important to report also FP and FN rates, or perhaps a combined measure such as F1. This is important given the expected class imbalance in most datasets. For example, the authors only reported the accuracy of 93% on the data shown in Figure 2a, although there is

clearly a large class imbalance, which makes the accuracy a misleading measure of performance.

- From Figure 1c and 1d, it seems that Cell Blast, compared to the other methods is very good at rejecting negative examples while it is comparable to the others in terms of the TP rate. Will be interesting to discuss this further.
- I find the comparison to scmap and CellFishing.jl in predicting continuous cell differentiation a bit unfair. A better comparison can be made to other tools dedicated to probabilistic prediction, such as: scANVI or fateID.
- The manuscript lacks a proper discussion section. The authors should adequately discuss the different aspects of model design and how it compares to other similar methods (e.g. scVI). How does the performance vary across methods per dataset (based on Figure 1d)? What are the important considerations which users and developers of querying methods should be aware of? What are the limitations of the current method? Also, a discussion of how they see their reference data evolving and how can their pretrained model adapt.

Minor comments

- Supplementary Table 2 can be enhanced by adding more details about the datasets, such as: number of cells and genes.
- Please specify throughout the manuscript whether scmap-cell or scmap-cluster was used.
- The figures and panels order do not always match the order by which they are mentioned in the manuscript.
- The y-axis label of Figure 1d is "Value". Please specify.
- In Figure 1d, the average value for scmap is lower than the numbers for positive and negative. Please explain.
- What is meant by "Time per query" in Figure 1e? Is it time per cell or per dataset? In case it is the later, I would argue that it is negligible since we are talking about a 300ms range.
- Please explain the Seurat Alignment score briefly.
- It is not clear in which experiments was the "online tuning" used. Please specify explicitly.
- Some details are missing from the methods section. For example, how was the feature selection performed, how many features were retained for each dataset, and how sensitive is Cell Blast to these choices?

Detailed Responses to Referees' Comments

Referee 1 Comments

The paper presents cell BLAST, a deep generative model to automatically identify cell types by searching a reference database. As more cells are sequenced and made available to the public, Cell BLAST (and similar tools) could be useful for us to rapidly interpret scRNA-seq results. The paper is well written and easy to follow.

Thanks for the encouraging comment!

I have some major concerns with the method and the results, which need to be addressed.

a) Since 2017, deep generative models have been used and established as valuable for different scRNA-seq data analysis tasks, e.g., dimension reductions, batch correction, and cell type annotations. In this manuscript, the authors use Adversarial Autoencoders (AAE), which is more flexible than variational autoencoders (VAE) but has more parameters and harder to train. In this manuscript, I don't see the advantages of using AAE to address any specific shortcomings of VAE in scRNA-seq data analysis. In several benchmark tests, e.g., dimension reductions, scVI (based on VAE) using a 5D latent space performs similarly to Cell Blast using 10D latent space (Supp. Fig.2).

Thanks for the insightful comment!

One key advantage of using AAE is that, instead of being limited to Gaussian distribution with diagonal covariance matrices (as in typical VAEs), the posterior distribution in AAE is effectively parameterized by the encoder neural network and can be learned from data directly. Particularly, such variational posterior reflects the “shape” of local data manifold (see Line 89-104 Page 5, and **Supplementary Fig. 6-7** for more details), enabling a novel “manifold-aware” similarity metric (NPD). Multiple assessments showed that the new metric is more accurate and stable than Euclidean distance in measuring cell-to-cell similarity (in the latent space, **Supplementary Fig. 6g, h; 8c**).

Meanwhile, as demonstrated in the **Methods** section (Line 396-402, Page 17), our model is rather stable to train despite the use of adversarial mechanisms. Intuitively, the cell embeddings in Cell BLAST lie in a low-dimensional latent space mapped from high-dimensional (gene) expression space, thus the support of cell embedding distribution and

prior distribution “span the full embedding space”, effectively extricating the model from the “disjoint support” problem, which is believed to be the major contributing factor to training instability in conventional GANs¹.

Last but not the least, while, as being pointed out correctly by the referee, AAE has more parameters than VAE due to the additional discriminator networks, the number of additional parameters is rather marginal, given the fact that the majority of parameters in autoencoder-based networks like AAE and VAE come from the encoder (especially the first layer) and the decoder (especially the last layer). For example, the table below lists the exact number of parameters in architecturally equivalent VAE-based and AAE-based networks for a typical use case with 1,000 genes as input, 10D latent embedding, and one 128D hidden layer in encoders, decoders and discriminators (bias terms are left out):

Layers	VAE	AAE
Encoder layer 1 (input to hidden layer)	1,000 * 128	1,000 * 128
Encoder layer 2 (hidden layer to embedding)	128 * 10 * 2	128 * 10
Decoder layer 1 (embedding to hidden layer)	10 * 128	10 * 128
Decoder layer 2 (hidden layer to output)	128 * 1,000 * 2	128 * 1,000 * 2
Prior discriminator layer 1 (embedding to hidden layer)	N.A.	10 * 128
Prior discriminator layer 2 (hidden layer to prediction)	N.A.	128 * 1
Total	387,840	387,968

It’s clearly shown that the AAE-based network contains just 128 more parameters (0.033% out of 387,840) than the VAE-based one. Meanwhile, Cell BLAST also supports categorical latent variable (notated as c in the manuscript) and adversarial batch alignment in addition to the basic configuration listed above, further leading to $128 * 20 + 20 * 10$ (= 2,760, or 0.71% increase, for categorical latent variable with 20 categories) + $10 * 128 + 128 * 10$ (= 2,560, or 0.66% increase, for adversarial batch alignment with 10 batches to be aligned). Thus, we believe that the additional ~1.38% parameter burden caused by adversarial components is minor and largely outweighed by its benefits.

Lastly, as described in the manuscript (Line 515-519, Page 22), the latent dimensionality (5D) of scVI in **Supplementary Fig. 3b, c** was selected based on its performance only. But we’d also like to clarify that reduced latent space dimensionality does not significantly affect the calculation above, as a VAE network with 5D latent space contains $1,000 * 128 + 128 * 5 * 2 + 5 * 128 + 128 * 1,000 * 2 = 385,920$ parameters, only ~0.5% less than a 10D latent space VAE.

b) I appreciate the authors put their tool online for people to use. A recent paper (PMID: 31500660) systematically compared 22 algorithms and found off-the-shelf SVM classifier works very well in practice. Cell BLAST did not perform so well compared to several classifiers such as SVM, RF, and LDA. Moreover, even scVI performed better than Cell BLAST in general. Finally, Cell BLAST is time-consuming to run. From this comparison paper, I don't see much of the values of using Cell Blast in practice. I do think that the authors should comment on and ideally repeat the Cell Blast experiments in the comparison paper.

Thanks for the reminder. After checking the released script by Abdelaal *et al.* ² (https://github.com/tabdelaal/scRNAseq_Benchmark), we found several flaws which, unfortunately, changed the standard behavior of Cell BLAST artificially and further led to misperception, such as 1) limiting model training process to 50 epoch only (instead of the default 1,000 epochs with dynamic early stopping), resulting in significant under-training for several datasets; 2) ignoring the feature selection step, increasing not only training difficulty but also computation time; 3) removing p-value-based hit filtering before making predictions, further confounding final prediction with unreliable hits.

After correcting these flaws (as well as various coding bugs), we re-ran the benchmark with exactly the same datasets (our full modified benchmark repository is available at https://github.com/gao-lab/scRNAseq_Benchmark), and found that the performance of Cell BLAST is among top tier, significantly better than scVI^a, RF, LDA, and comparable with SVM (**Additional Fig. 1-2**).

^a After close inspection, we found that the Abdelaal *et al* did not used the original unsupervised scVI but rather the extended supervised scANVI model³.

Additional Figure 1 Repeating the intra-dataset and inter-dataset experiments in Abdelaal *et al.* using Cell BLAST with / without online tuning. (a) Median F1-score of intra-dataset predictions (corresponding to Abdelaal *et al.*'s Fig. 1a). **(b)** Percent of unlabeled cells in intra-dataset predictions (corresponding to Abdelaal *et al.*'s Fig. 1b). **(c)** Median F1-score of inter-dataset predictions in the "PbmcBench" datasets (corresponding to Abdelaal *et al.*'s Fig. 3a). **(d)** Median F1-score of inter-dataset predictions in the brain datasets, with major lineage annotation (corresponding to Abdelaal *et al.*'s Fig. 4a). **(e)** Median F1-score of

inter-dataset predictions in the brain datasets, with deeper level annotation of 34 cell populations (corresponding to Abdelaal *et al.*'s **Fig. 4b**). **(f)** Median F1-score of inter-dataset predictions in the “CellBench” datasets (corresponding to Abdelaal *et al.*'s **Fig. S8a**). **(g)** Median F1-score of inter-dataset predictions in the pancreatic datasets (corresponding to Abdelaal *et al.*'s **Fig. 5a**).

Additional Figure 2 Repeating the rejection experiments in Abdelaal *et al.* using Cell BLAST with / without online tuning. (a) Percent of unlabeled cells in the negative control experiment (corresponding to Abdelaal *et al.*'s **Fig. 6a**). **(b)** Percent of unlabeled cells in the unseen population experiment (corresponding to Abdelaal *et al.*'s **Fig. 6b**).

We failed to repeat the computation time assessment in Abdelaal *et al.*, due to the unavailability of benchmarking hardware configuration. However, we found that Cell BLAST took ~1.3 min for the “Xin” dataset (instead of the 75 min as stated in Abdelaal *et al.*) and just ~8.6 min for the largest one (“Zheng 68K”) on a standard Linux server in CPU-only mode (Intel Xeon Gold 6240 CPU, 192G RAM, limited to 8 threads per job to simulate typical desktop computation power). As comparison, SVM took ~4.2 min, SingleCellNet took ~2.5 h and SingleR took ~6.3h on the largest “Zheng 68K” dataset with the same feature set and hardware configuration.

Last but not the least, we’d like to point up the fact that Cell BLAST is more than a dedicated cell typing tool but also an accurate and robust general-purpose querying algorithm for heterogeneous single-cell transcriptome datasets (Line 25-26 Page 3, Line 260-266 Page 10, also see Response to Comment 3 of the 2nd referee for more discussion). The unsupervised nature of Cell BLAST model enables not only identifying novel type of cells (Line 151-172, Page 7) but also predicting various types of features including continuous cell differentiation potential (Line 197-218, Page 8-9).

c) From the experiments presented in this manuscript, it seems that Cell BLAST works well, and the p-values provided by Cell BLAST is useful for users to interpret the results. However, as deep learning models are 'black box' methods and are prone to adversarial attacks, extra information is helpful. For example, the authors can compute the ROCs of marker genes in discriminating a cell type from other cells.

Thanks for the valuable recommendation. To help users' interpretation, we designed and implemented a data-driven, gradient-based gene ranking algorithm to identify marker genes for particular cell types. Briefly, we used each cell type in turn as query data and the remaining cell types as reference data to perform Cell BLAST querying. Then the embedding space deviation of query cells from their nearest reference hits are propagated back to the original gene space by computing the gradient of the encoder neural network. Genes with higher gradient values indicate that the model "sees" them as "marker genes" of the particular query cell type (see Line 132-143, Page 6, as well as the **Methods** section, Line 691-713, Page 27-28 for more details). GSEA analysis shows that manually annotated cell type markers in the PanglaoDB database⁴ are significantly over-represented in genes with higher gradient values for the corresponding cell types (**Supplementary Fig. 10**), suggesting its high consistency between the model's internal logic and prior biological knowledge.

The new algorithm has been integrated into the standalone Cell BLAST package (v0.3.6) and will be available online at the Web Server in few months.

d) The authors introduced an adversarial batch correction term in the objection function and a weight lambda parameter to balance its contribution to the objective function. This parameter could be data-dependent. In Fig.1, it seems that Cell Blast did a decent job in batch-correction for the specific dataset. However, we still see strong batch effects in other datasets (e.g., Supp. Fig. 3g, Supp. Fig. 4a). Therefore, to prove the robustness and effectiveness of this adversarial batch-correction strategy, the authors need to do more side-by-side comparisons (e.g., with scVI) across different datasets.

Thanks for the reminder. We agree with the referee that the hyperparameter λ_b can be data-dependent. However, our empirical evaluations suggest that the current default λ_b (0.01) works reasonably well across various datasets (**Additional Fig. 3**).

Additional Figure 3 MAP-SAS plot with different values of lambda. We took step-wise grid searching for a range of λ_b values from 0 to 1 for all four benchmark cases, and found that a λ_b value of 0.01 is always close to the optimal “elbow point” on the MAP-SAS plot, so we chose it as the default value for all adversarial alignment-based results in the manuscript as well as the online server.

On the other hand, it seems there is sort of misunderstanding for the two specific cases this referee referred to:

As for **Supplementary Fig. 4g** (originally **Supplementary Fig. 3g^b**), it visualizes the batch correction result of the *Tabula Muris* dataset (“Quake_10x” and “Quake_Smart-seq2”). According to the original published paper⁵, cells profiled by these two technologies differ in terms of cell type composition (e.g., pancreatic cells only exist in “Quake_Smart-seq2” but not in “Quake_10x”, while kidney loop Henle cells only exist in “Quake_10x” and “Quake_Smart-seq2”, see **Supplementary Table 3** (sheet 4) for all differences in cell type composition), so the two datasets are not supposed to be perfectly mixed *per se*.

Meanwhile, in **Supplementary Fig. 5a** (originally **Supplementary Fig. 4a**), we intentionally shows the fact that while canonical batch correction strategy works for cross-dataset batch effect (i.e. the batch effect observed among six independent datasets: “Baron_human”, “Enge”, “Lawlor”, “Muraro”, “Segerstolpe” and “Xin_2016”), it does not handle

^b As we inserted a new **Supplementary Fig. 2**, the previous **Supplementary Fig. 3-4** are now **Supplementary Fig. 4-5**.

within-dataset batch effects (i.e. the batch effect among four different donors within the same “Baron_human” dataset) correctly (also see **Additional Fig. 4** for the output produced by scVI). As such, we went on in the manuscript to demonstrate that the adversarial batch correction strategy employed by Cell BLAST can be extended to deal with multiple levels of batch effect simultaneously: as shown in **Supplementary Fig. 5d-h**, Cell BLAST successfully removed both cross-dataset and within-dataset batch effect when configured to do so (**Supplementary Fig. 5f** is the equivalent of **Supplementary Fig. 5a** with multi-level batch effect correction enabled).

Additional Figure 4 Within-dataset batch effect also remains when cross-dataset batch effect is corrected using scVI. This is the scVI equivalent of Supplementary Fig. 5a.

We’re sorry for such misunderstanding, and have revised the manuscript with these additional explanations accordingly (Legend of **Supplementary Fig. 4**, and Line 79-86, Page 4-5).

e) Currently, people have collected large numbers of scRNA-seq datasets, e.g., the panglaodb (<https://panglaodb.se/index.html>) has about 5.6 million collected and annotated cells. Moreover, the authors provide the marker genes for cell annotations. I think these marker genes could be useful for interpreting Cell Blast results (as in c).

Thanks for the helpful recommendation! We adopted the suggestion in the evaluation of our model interpretation method based on gene-space gradients. See the Response to Comment (c) for details.

f) The authors used a 'normalized projection distance' for cell query and shows it improves Cell Blast query but has almost no effects on scVI results. It's not clear to me why this is the

case. Have the authors do the comparisons on different datasets? How the weight (beta) of the KL divergence term in the VAE objective function influence the results?

Thanks for the comment. We have improved the section “A posterior-based cell-to-cell similarity metric” to better explain the idea behind NPD and the difference between Cell BLAST posterior and scVI posterior (Line 88-115, Page 5). Briefly, we believe that the observed disparity roots from the fact that the posterior distribution in AAE-based model (i.e. Cell BLAST) is effectively parameterized by the encoder neural network and can be learned from data directly, while the distribution in VAE-based model (e.g. scVI) is strictly limited to Gaussian distribution with diagonal covariance matrices (also see Response to Comment (a) for more). Such inaccurately modeled posterior distribution will simply result in more harm for the “manifold-aware” similarity metric (NPD) than for the canonical “manifold-blind” Euclidean distance. Consistently, we confirmed the idea empirically over multiple datasets (**Supplementary Fig. 6-7**).

Given that the diagonal covariance constraint of scVI posterior is hard-coded, adjusting the KL divergence weight would hardly make any improvement. To demonstrate that, we tested different KL divergence weights for scVI, and also different adversarial prior regularization weights for Cell BLAST. As expected, NPD consistently increased the performance of Cell BLAST, while decreased the performance of scVI regardless of KL regularization weight (**Supplementary Fig. 8c**).

In summary, I think that Cell BLAST is an interesting and useful tool, especially for people with limited computational skills. However, I'm not convinced that it significantly improves similar tools for automatic cell annotations and for robust batch corrections given extra parameters and the results presented in this paper and in the comparison paper(PMID: 31500660).

Referee 2 Comments

The manuscript by Cao et al. presents Cell Blast, a method and webtool to automatically assign cell identities by co-embedding single cell RNA-seq data with annotated reference data. The authors propose a generative model based on neural-networks to learn non-linear embeddings of high dimensional single cell RNA-seq data. The authors evaluated their method in terms of the accuracy of the resulting low dimensional embeddings, batch effect correction, and cell identification.

The proposed method is interesting and, in particular, the webtool and the curated reference data can be a valuable resource for the community.

Thanks for the encouraging comment!

However, the manuscript would benefit from addressing a number of issues in greater detail - see below.

Major comments

1. The paper can greatly benefit from a figure illustrating the structure of the generative model used by the authors to improve readability of the methods section.

Thanks for the valuable suggestion! We have added a corresponding figure illustrating the model structure (**Supplementary Fig. 1a**).

2. The model presented in this paper is similar to scVI. The major difference appears to be the reliance on adversarial training between the encoder and the discriminator networks while scVI relies on variational inference. In that sense, a proper discussion of the similarities and differences between the proposed model and scVI is important.

Thanks for the valuable suggestion, and we have appended a dedicated discussion into the manuscript accordingly (Line 99-115, Page 5).

Briefly, both scVI and Cell BLAST use variational inference, and the key difference lies in the fact that the canonical variational autoencoder model used by scVI enforces Gaussian distribution (with diagonal covariance matrices) for the variational posterior distribution, while the adversarial component in Cell BLAST enables a free form of variational posterior

distribution which is learned from data directly (via the encoder neural network). Multiple assessments have demonstrated that Cell BLAST models variational posterior more accurately than scVI, and that the improved variational posterior is necessary for accurate cell querying (**Supplementary Fig. 6-7, Supplementary Fig. 8c**, also see Response to Comment (a) and (f) of the first referee for more details on model comparison).

3. I disagree with the authors regarding the ability of scVI to perform cell querying (Supplementary Table 1). A modified version of scVI (scANVI) allows annotation of cells based on the posterior distribution, much like Cell Blast. This is described in a preprint: <https://www.biorxiv.org/content/10.1101/532895v1> and on GitHub: <https://nbviewer.jupyter.org/github/YosefLab/scVI/blob/master/tests/notebooks/annotation.ipynb>.

As such, I believe a proper comparison of both methods in terms of cell querying is important.

Thanks for the suggestion. In our manuscript, we defined cell querying as the process of unsupervisedly finding the reference cells most similar to the query cells (Line 26, Page 3), similar to searching the internet via a search engine like Google. The output of Cell BLAST (and other cell querying methods like scmap and CellFishing.jl) is a list of similar reference cells along with corresponding similarity metrics, so that existing annotations (like cell type and cell differentiation potential) in curated references can be utilized to annotate newly sequenced cells based on the identified transcriptomic similarity. Notably, analogous to classical biological sequence analysis, various similarity-based downstream analyses could be done, including but not limited to cell typing (e.g. Line 151-172, Page 7 for identifying novel type of cells, Line 196-218, Page 8-9 for predicting continuous cell differentiation potential).

scANVI is designed and implemented based on a semi-supervised classification model, which does not fit our definition of cell querying algorithm. In particular, it does not return a list of most similar reference cells and similarity values that can be used to reject query cells like in other cell querying methods. Thus, we tested scANVI in our cell querying benchmark with the outputted maximal classification probability as a proxy of similarity to the predicted cell type. Unfortunately, it did not perform well compared to other methods whether or not dataset normalization is applied^c (**Additional Fig. 5**).

^c Like scVI, scANVI is supposed to be trained on raw count data. However, scANVI model trained on raw count data may not generalize well in the cell querying setup, where it is applied to unseen test datasets that can have different sequencing depths. As such, we tried scANVI with both raw count (default setting) and normalized count (intuitively more generalizable) data.

Additional Figure 5 Cell querying benchmark with scANVI incorporated. (a) MBA curves reflecting the balance between predicting positive cell types and rejecting negative cell types (equivalent of **Supplementary Fig. 9a**). (b) AUC (Area Under Curve) of the MBA curves in (a) (equivalent of **Supplementary Fig. 9b**). (c) Average MBA at different cutoff values (equivalent of **Supplementary Fig. 9c**). (d) Positive and negative cell type MBA under the optimal cutoff of each method (equivalent of **Fig. 1c**). (e) Positive, negative and average cell type MBA under the optimal cutoff of each method (equivalent of **Fig. 1d**).

4. *One of the strong points of the paper is the extensive curated collection of reference data. However, the rapid growth of scRNA-seq data will deem any reference outdated very quickly. What are the authors thoughts regarding maintaining their curated reference atlas? And more importantly, how does updating the atlas affect the pretrained model?*

Thanks for the encouraging comment. Aiming to be a high-quality multispecies reference database, ACA is under regular update with the established protocol described in the **Methods**. Briefly, we regularly check new released datasets in NCBI GEO, and curate the raw expression data based on the publications as well as meta-data provided by the original authors. Meanwhile, inspired by the classic Blast2GO⁶ algorithm, we also implement a query-based, ontology-aware inference strategy for assigning Cell Ontology terms (BLAST2CO) to unlabeled data during curation. Empirical case study shows that both the incorporation of hit similarity and ontology structure lead to improved Cell Ontology assignment, especially for cells marked as “ambiguous” by the standard major voting strategy (**Supplementary Fig. 19**, also see Line 273-285, Page 10-11 for more details).

Depending on the size and cell type composition of newly acquired data, existing models may be either fine-tuned (via the “online-tuning” strategy, see Line 631-643, Page 25-26 for more details) or built from scratch. Since ACA’s first public release in early 2019, the database has been updated with 5 new datasets (for 5 organs), with 1 model fine-tuned and 5 models built from scratch.

We have added last updated date as well as the update strategy for all ACA references online.

5. *I got confused in the explanation of the stochastic autoencoder part (equations 6 – 9). In particular, how is it possible to skip the Poisson sampling. Please explain.*

Thanks for the meticulous question. During model training and variational posterior sampling, we apply equation 6-9 in order. When obtaining the point estimate of the variational posterior, we simply replace equation 7 with $\tilde{\mathbf{x}} = \hat{\mathbf{x}}$ (i.e., skip the Poisson sampling), and apply the other equations as is:

$$\hat{\mathbf{x}} = \frac{10^4 \cdot \mathbf{x}}{\sum_{j \in G} \mathbf{x}_j} \quad (6)$$

$$\tilde{\mathbf{x}} = \hat{\mathbf{x}} \quad (7')$$

$$\mathbf{z} = \text{Enc}_z(\tilde{\mathbf{x}}; \phi_{\text{Enc}}), \mathbf{c} = \text{Enc}_c(\tilde{\mathbf{x}}; \phi_{\text{Enc}}) \quad (8)$$

$$\mathbf{l} = \mathbf{z} + H\mathbf{c} \quad (9)$$

Programmatically, it is implemented by feeding the normalized data directly to the tensor $\tilde{\mathbf{x}}$ of Poisson sampling output, effectively “shortcutting” the Poisson sampling operation:

https://github.com/gao-lab/Cell_BLAST/blob/fa1f30d2d54b68a06479513164746a80c1fdb031/Cell_BLAST/directi.py#L415).

6. The authors rely on “Cell type nearest neighbor mean average precision (MAP)” to evaluate the accuracy of dimensionality reduction. This measure only evaluates how homogenous are the resulting low dimensional embeddings, with no regard to the correspondence of the neighborhood structure between the high and low dimensional spaces. My concern here stems from the fact that most of the cell labels are derived from manual annotations based on overlaying marker gene expression on low dimensional embeddings. Perhaps the authors should consider a more appropriate measure, such as:
<https://www.biorxiv.org/content/10.1101/786269v1>

Thanks for the insightful comment! While preserving (finer) structures like within-cell-type variation during dimensionality reduction is strongly desired, practically, a fundamental challenge for performance assessment in real-world is the lack of a “reference” distance metric that characterizes the “genuine” functional variation of interest, or what distance metric is “appropriate” for the original high-dimensional space. For example, the commonly used Euclidean metric assumes inter-feature independence which is largely unrealistic for scRNA-seq data, especially given the fact that functionally related genes are usually expressed with high correlation^d. Moreover, common distance metrics applied to the high-dimensional gene expression space are also confounded by data corruptions like dropout and measurement noise in scRNA-seq data.

The MCV paper⁷ suggested by the referee is an interesting attempt regarding the evaluation of scRNA-seq data imputation methods. However, not all dimension reduction methods are trained by data reconstruction (e.g., some methods like tSNE are distance-based). It seems unclear how a similar strategy can be adapted to evaluate these dimension reduction methods, since it does not address the lack of a “reference” metric space. As such, with no absolutely appropriate “reference” evaluation metric, we believe that metrics based on the phenotypic evidence of cell types (including MAP) are a reasonable compromise, and have been adopted by many other works on dimension reduction methods^{8, 9, 10, 11}.

^d Ideally, distance metrics aimed at characterizing the “cellular function space” should collapse functionally correlated genes into one feature (e.g., a functional module), instead of “overcounting” their contribution.

7. I am curious why did the authors decide to exclude tSNE and UMAP from the comparison of dimensionality reduction methods, while both are widely used within the single cell community.

Thanks for the recommendation. We did not include these methods previously as they are primarily used for data visualization. As suggested, we have now incorporated them into the dimensionality reduction benchmark, and found that they performed rather comparable with Cell BLAST in the benchmark datasets (**Supplementary Fig. 3**).

8. While comparing the batch effect correction, the authors used PCA to obtain low dimensional embeddings for the corrected data of ComBat, MNN, and CCA anchor. This seems a bit unfair given that PCA on its own did not perform well, based on Figure 1b. Why not calculate MAP for these methods on the high dimensional data directly?

Thanks for the comment. Accordingly, we repeated the assessment with high dimensional data-based results incorporated, and found increased Seurat alignment scores, at the cost of lower Mean average prevision (i.e. cell type resolution) (**Additional Figure 6**).

Meanwhile, PCA is essentially a “batch-blind” dimension reduction algorithm as it simply tries to preserve the most prominent data variations, regardless of their source[°]. Given 1) the distinct representation of batch-corrected data among various methods (ComBat, MNN, and CCA anchor in high dimensional space while CCA, scVI, and Cell BLAST in dimension-reduced latent space), and 2) the various challenges for precisely modeling distance for high-dimensional scRNA-seq data (see Response to Comment 6 for more details), we believe that the “batch-blind” PCA operation helps a fair comparison.

[°] Which also makes PCA a proper “negative control” for embedding-based batch correction.

Additional Figure 6 MAP-SAS plot with high dimensional data-based results incorporated. Corresponding to **Fig. 1b** in the manuscript. Notably, the fact that Seurat alignment score decreases after PCA for ComBat, MNN and CCA anchor indicates that batch effect is still one of the major sources of variation in the batch-corrected expression matrices.

9. When comparing the performance of cell querying (Figure 1c and 1d), the authors report only the TP and TN values. It is important to report also FP and FN rates, or perhaps a combined measure such as F1. This is important given the expected class imbalance in most datasets. For example, the authors only reported the accuracy of 93% on the data shown in Figure 2a, although there is clearly a large class imbalance, which makes the accuracy a misleading measure of performance

Thanks for the reminder on the important issue! Given the class imbalance here, we have replaced the original plain accuracy with Mean Balanced Accuracy (MBA) metric for **Fig. 2a** and the tracheal case (Line 155, Page 7, also see Equation 29 for more details). Essentially, positive cell type MBA not only requires that the positive cells are accepted (not rejected), but also that the cells are given the correct cell type prediction. Negative cell type MBA requires that the negative cells are either rejected, or predicted as a limited set of closely-related cell types (as specified in **Supplementary Table 5**). We also reported the combined metric of average MBA (third group in **Fig. 1d**, see Line 647-671, Page 26-27 for definition), which is an analogue to the F1 score.

Moreover, we'd clarify that it is the positive and negative cell type MBA values that are reported in **Fig. 1c, d**, which should not be confused with TP and TN values.

10. From Figure 1c and 1d, it seems that Cell Blast, compared to the other methods is very good at rejecting negative examples while it is comparable to the others in terms of the TP rate. Will be interesting to discuss this further.

Thanks for the suggestion. Cell BLAST is designed as a general-purpose cell querying algorithm for searching large-scale scRNA-seq database. Analogous to the classic BLAST algorithm for sequence analysis, effectively excluding false hits is essential for its mission. In addition to the model design, we believe that the novel posterior-based, “manifold-aware” similarity metric effectively contributes to the performance improvement. Intuitively, the latent-space posterior density is expected to decrease more steeply between query and false hits than positive hits because of the lower reconstruction likelihood $p(\mathbf{x}_{ref}|\mathbf{l}_{query})$ and $p(\mathbf{x}_{query}|\mathbf{l}_{ref})$, while the popular Euclidean distance metric is posterior-blind and is more likely to make false-positive calls (also see Line 89-115, Page 5 for more detailed discussions and empirical assessments).

11. I find the comparison to scmap and CellFishing.jl in predicting continuous cell differentiation a bit unfair. A better comparison can be made to other tools dedicated to probabilistic prediction, such as: scANVI or fateID.

Thanks for the suggestion.

scANVI³ is designed as a semi-supervised classification model, accepting discrete cell class labels as supervision and returning continuous class distribution for the a given transcriptome. Thus, as being suggested by the referee, we selected reference cells with maximal lineage probability > 0.5 as the labelled training set and took the output class distribution as the lineage differentiation probabilities. The batch effect correction function is also enabled to correct the batch effect between reference and query data. As shown in **Supplementary Fig. 16b, g**, and **Fig. 2f**, scANVI did not perform well. Specifically, **Supplementary Fig. 16g** reveals that the class distributions predicted by scANVI are almost one-hot and failed to model the continuous differentiation process correctly. Meanwhile, scANVI aligned many unrelated query cells to erythrocyte progenitors in the reference dataset, suggesting an inaccurate batch effect correction.

FateID¹² is a method similar to PBA, which infers differentiation probabilities from a dataset, under manual supervision in terms of terminal cell fates. The fact that FateID cannot be trained on a reference dataset and applied to another dataset for prediction excludes it from current comparison theme.

12. The manuscript lacks a proper discussion section. The authors should adequately discuss the different aspects of model design and how it compares to other similar methods (e.g. scVI). How does the performance vary across methods per dataset (based on Figure 1d)? What are the important considerations which users and developers of querying methods should be aware of? What are the limitations of the current method? Also, a discussion of how they see their reference data evolving and how can their pretrained model adapt.

Thanks for the valuable suggestion! We have revised the discussion section accordingly (Line 248-287, Page 10-11).

Minor comments

13. Supplementary Table 2 can be enhanced by adding more details about the datasets, such as: number of cells and genes.

Thanks for the suggestion. We have modified the table as suggested (**Supplementary Table 2**).

14. Please specify throughout the manuscript whether scmap-cell or scmap-cluster was used.

Thanks for the comment. We have used scmap-cell throughout the manuscript, because only scmap-cell falls into our definition of “cell querying” (see Response to Comment 3 for details). A specific statement has been added to the **Methods** section (Line 673-674, Page 27).

15. The figures and panels order do not always match the order by which they are mentioned in the manuscript.

Thanks for the suggestion. It appears that the problem is most prominent in the ionocyte case study. We have extensively rearranged **Supplementary Fig. 11-15** to meet the order they are mentioned in the manuscript. Yet there are still some minor discrepancies in figure order. E.g., **Supplementary Fig. 6g, h** and **Supplementary Fig. 7g, h** are mentioned after **Supplementary Fig. 8a**, but the dataset correspondence would be much clearer if they are grouped with other figures in **Supplementary Fig. 6** and **Supplementary Fig. 7**. As such, we prefer to keep the current order in this specific case.

16. The y-axis label of Figure 1d is “Value”. Please specify.

Thanks for the suggestion. The y-axis label “Value” should be “Mean average precision” (which was incorrectly placed in x-axis label). We have fixed the axis labels as suggested (**Fig. 1d**).

17. In Figure 1d, the average value for scmap is lower than the numbers for positive and negative. Please explain.

Thanks for the comment. The middle lines in the boxplot are medians rather than means. It is indeed possible that the median of the average of two variables is lower than the median of each variable. For the record, the MBA values for scmap in **Fig. 1d** are listed below:

Experiment	Positive types	Negative types	Average
1	0.841	0.760	0.801
2	0.818	0.830	0.824
3	0.654	0.885	0.770
4	0.896	0.507	0.702
5	0.846	0.758	0.802
6	0.818	0.827	0.823
7	0.638	0.836	0.737
8	0.899	0.507	0.703
9	0.843	0.728	0.786
10	0.815	0.828	0.821
11	0.629	0.837	0.733
12	0.906	0.528	0.717
13	0.841	0.748	0.794
14	0.806	0.834	0.820
15	0.653	0.856	0.754

16	0.899	0.531	0.715
Median	0.830	0.794	0.778

18. What is meant by “Time per query” in Figure 1e? Is it time per cell or per dataset? In case it is the later, I would argue that it is negligible since we are talking about a 300ms range.

Thanks for the reminder! The “Time per query” should be read as “Time per (query) cell”, so the time difference is quite substantial. We have revised the y-axis label of **Fig. 1e** to be “Time per query cell” to avoid possible confusion.

19. Please explain the Seurat Alignment score briefly.

Thanks for the suggestion. We have added a brief explanation for Seurat alignment score in the **Methods** section (Line 534-541, Page 22).

20. It is not clear in which experiments was the “online tuning” used. Please specify explicitly.

Thanks for the comment. We used “online tuning” only in the cross-species querying between the “Tusi” and the “Velten” dataset in the hematopoiesis case study. Specific statements can be found in Line 213 Page 8, Line 643-645 Page 26, and Line 762 Page 30.

21. Some details are missing from the methods section. For example, how was the feature selection performed, how many features were retained for each dataset, and how sensitive is Cell Blast to these choices?

Thanks for the comment. We used the “FindVariableGenes” function in Seurat v2 as the feature selection method. We have added some detailed explanation in the **Methods** section (Line 480-482, Page 20). The number of features retained for each dataset has been added to **Supplementary Table 2**. We also added a feature selection benchmark to evaluate the robustness of Cell BLAST to different feature selection thresholds (**Supplementary Fig. 2**). We found that the performance of Cell BLAST is relatively stable as the number of selected

genes varies from 500-5,000. Gene sets with < 500 genes contain insufficient information while gene sets with $> 5,000$ genes contain too much noise and also make the model difficult to train (especially for small datasets like “Muraro” and “Adam”).

References

1. Arjovsky M, Bottou L. Towards Principled Methods for Training Generative Adversarial Networks. *arXiv e-prints*, (2017).
2. Abdelaal T, *et al.* A comparison of automatic cell identification methods for single-cell RNA sequencing data. *Genome Biol* **20**, 194 (2019).
3. Xu C, Lopez R, Mehlman E, Regier J, Jordan MI, Yosef N. Harmonization and Annotation of Single-cell Transcriptomics data with Deep Generative Models. *bioRxiv*, 532895 (2019).
4. Franzen O, Gan LM, Bjorkegren JLM. PanglaoDB: a web server for exploration of mouse and human single-cell RNA sequencing data. *Database (Oxford)* **2019**, (2019).
5. Tabula Muris C, *et al.* Single-cell transcriptomics of 20 mouse organs creates a Tabula Muris. *Nature* **562**, 367-372 (2018).
6. Conesa A, Gotz S, Garcia-Gomez JM, Terol J, Talon M, Robles M. Blast2GO: a universal tool for annotation, visualization and analysis in functional genomics research. *Bioinformatics* **21**, 3674-3676 (2005).
7. Batson J, Royer L, Webber J. Molecular Cross-Validation for Single-Cell RNA-seq. *bioRxiv*, 786269 (2019).
8. Alavi A, Ruffalo M, Parvangada A, Huang Z, Bar-Joseph Z. A web server for comparative analysis of single-cell RNA-seq data. *Nat Commun* **9**, 4768 (2018).

9. Lopez R, Regier J, Cole MB, Jordan MI, Yosef N. Deep generative modeling for single-cell transcriptomics. *Nat Methods* **15**, 1053-1058 (2018).
10. Becht E, *et al.* Dimensionality reduction for visualizing single-cell data using UMAP. *Nat Biotechnol*, (2018).
11. Li X, Chen W, Chen Y, Zhang X, Gu J, Zhang MQ. Network embedding-based representation learning for single cell RNA-seq data. *Nucleic Acids Res* **45**, e166 (2017).
12. Herman JS, Sagar, Grun D. FateID infers cell fate bias in multipotent progenitors from single-cell RNA-seq data. *Nat Methods* **15**, 379-386 (2018).

Reviewers' Comments:

Reviewer #1:

Remarks to the Author:

The authors are responsive in addressing my comments and thus the paper has been improved. I have a few further comments after reading the letter and the revised manuscript, mainly about the experiments and results.

- a) First, I appreciate the authors for comparing the parameters for VAE and AAE and also the experiments demonstrating that Cell Blast is among the top-tier methods. The authors claim that Cell Blast is stable and easy to train, but it seems that it took a quite large number of epochs (1000 epochs). It's good to compare the training dynamics with those of VAE-based methods to show training is stable.
- b) The authors pointed out that feature selection is key to get performance on par with SVM. In the revision, the authors also showed Cell blast results were robust to the number of selected genes. However, there are potential problems for selecting genes for querying cells. For example, some novel cells may only express a few selected genes. This is especially problematic for large datasets as many more cells are becoming available. So potentially a large number of genes should be used for a large reference dataset. How many genes are sufficient to separate the cell types in the reference dataset? The gene selection section of the paper is unclear to me: for a large reference dataset including cells from different studies, do the authors use the union of genes selected in 50% datasets? Will this approach filter out marker genes of rare cell types?
- c) Batch correction is important for effective querying cells. In the revision, the authors clarified and emphasized in Supp. Fig.5 that Cell Blast can deal with both inter-dataset and intra-dataset batch effects. However, this approach seems to be limited as you need to train an extra NN for each added batch vector. Instead, the Harmony tool has been developed for this task. Besides, a recent study (PMID: 31948481) showed that Harmony is generally the best batch-correction method for scRNA-seq data but missed in the benchmarking. The authors need to compare to the state-of-the-art to show it's effectiveness. Also, does Harmony + other methods (e.g., PCA) do better than Cell Blast? More recently, a related method scsphere also based on NN, claimed to be able to deal with multiple batch effects. The SAUCIE method, also based on NN was able to correct batch effects.
- d) For benchmarking dimension reduction, a more proper way is to use the same number of dimensions, e.g., 10 (although for linear methods such as PCA need more PCs). Also, it's good to add the parameters for different methods as DM methods such as tSNE are sensitive to parameter settings.
- e) The authors showed the Smart-seq2 data results in additional Fig.3. It's not clear the zero-inflation in Smart-seq2 data influence the results (Cell Blast used Gamma-Poisson distribution for the count data)?
- f) Should doublets be removed first for using Cell blast, or cell blast is sensitive to annotate doublets?

Reviewer #2:

Remarks to the Author:

I would like to thank the authors for adequately addressing all the comments. I believe the manuscript has improved significantly.

The only additional remark I have is that I think Supplementary Fig1A deserves to be in the main

text to help the readers with the model interpretation. After all, this is a methods paper and I find it strange that all the method description goes to the supplementary materials.

Detailed Responses to Referees' Comments

Referee 1 Comments

The authors are responsive in addressing my comments and thus the paper has been improved. I have a few further comments after reading the letter and the revised manuscript, mainly about the experiments and results.

a) First, I appreciate the authors for comparing the parameters for VAE and AAE and also the experiments demonstrating that Cell Blast is among the top-tier methods. The authors claim that Cell Blast is stable and easy to train, but it seems that it took a quite large number of epochs (1000 epochs). It's good to compare the training dynamics with those of VAE-based methods to show training is stable.

Thanks for the encouraging comment and reminder. A direct comparison of negative log-likelihood for scVI and Cell BLAST while being trained on the “Baron_human” dataset¹ reveals similar training dynamics (**Supplementary Fig. 1a**), triggering early-stopping at 218 and 173 epochs respectively*. Of note, the losses of Cell BLAST adversarial component also converge stably during training (**Supplementary Fig. 1b**), as they operate on low-dimensional cell embeddings mapped from high-dimensional gene expression data (see Line 404-410, Page 18-19 for details).

b) The authors pointed out that feature selection is key to get performance on par with SVM. In the revision, the authors also showed Cell blast results were robust to the number of selected genes. However, there are potential problems for selecting genes for querying cells. For example, some novel cells may only express a few selected genes. This is especially problematic for large datasets as many more cells are becoming available. So potentially a large number of genes should be used for a large reference dataset. How many genes are sufficient to separate the cell types in the reference dataset? The gene selection section of the paper is unclear to me: for a large reference dataset including cells from different studies, do the authors use the union of genes selected in 50% datasets? Will this approach filter out marker genes of rare cell types?

* The default number of maximal training epochs is set to 1,000 in the current Cell BLAST Python package, with early-stopping technique employed which will stop the training process as soon as loss function on independent validation data no longer decreases.

Thanks for the insightful comment.

As pointed out by the referee, Cell BLAST is reasonably robust to the number of selected genes according to our feature selection benchmark (**Supplementary Fig. 1d**). Cell type resolution is roughly constant as the number of selected genes range from 500 to 5,000 in our benchmark datasets. Intuitively, the number of genes required to separate cell types in the reference dataset would increase with the level of data heterogeneity rather than data size. While it is hardly practical to give an exact number considering the various cell composition among different reference data, the default cutoffs work pretty well when training models on ACA reference panels, with all distinct cell types separated at least as well as in the original publication. Meanwhile, we also provide a guideline including sanity check for users training models on custom reference data (https://cblast.gao-lab.org/doc-latest/_static/DIRECTi.html).

In the manuscript, we used the union of genes selected in 50% of “batches” when conducting gene selection for individual datasets (to mitigate within-dataset batch effect). Since the cell type composition of different batches within the same dataset is usually similar, the 50% cutoff should not cause missing markers[†]. When merging multiple datasets, we took the union of genes selected in individual datasets. Thus, markers of rare cell types existent in only one dataset would also be included. We have restructured related method sections to avoid further confusion [Line 501-519, Page 22].

Meanwhile, as what being well demonstrated by the ionocyte case study presented in the manuscript, the non-linear NN-based model employed by Cell BLAST is able to identify rare novel cells even without the inclusion of their own markers[‡].

c) Bath correction is important for effective querying cells. In the revision, the authors clarified and emphasized in Supp. Fig.5 that Cell Blast can deal with both inter-dataset and intra-dataset batch effects. However, this approach seems to be limited as you need to train an extra NN for each added batch vector. Instead, the Harmony tool has been developed for this task. Besides, a recent study (PMID: 31948481) showed that Harmony is generally the best batch-correction method for scRNA-seq data but missed in the benchmarking. The

[†] Meanwhile, we have made the 50% cutoff an easily tunable parameter [https://cblast.gao-lab.org/doc-latest/modules/Cell_BLAST.data.html#Cell_BLAST.data.ExprDataSet.find_variable_genes], which users can modify if different batches vary considerably in cell type composition.

[‡] In the ionocyte case, where we reselected genes after the removal of ionocytes from reference data, there is indeed no ionocyte marker in the selected genes, and the model is still able to identify querying ionocytes as a novel cell type (**Fig. 3b-c**). This is because, intuitively, different expression pattern in non-marker genes is also informative, e.g., the lack of expression of reference cell type markers is in itself an indication of novel cell types.

authors need to compare to the state-of-the-art to show it's effectiveness. Also, does Harmony + other methods (e.g., PCA) do better than Cell Blast? More recently, a related method scSphere also based on NN, claimed to be able to deal with multiple batch effects. The SAUCIE method, also based on NN was able to correct batch effects.

Thanks for the reminder. As being suggested by the referee, we have added scSphere², Harmony³, and SAUCIE⁴ to the batch effect correction benchmark (**Fig. 2a**)[§]. scSphere compared unfavorably with Cell BLAST in terms of both MAP and Seurat alignment score, and applying scSphere to our multilevel batch correction experiment reveals that scSphere achieved lower cell type resolution and lower batch correction performance for all levels of batch effects (**Additional Fig. 1**). On the other hand, Harmony was able to preserve true biological signal similarly well, but achieved lower batch alignment performance compared to Cell BLAST. Moreover, SAUCIE tends to overalign and achieved significantly lower MAP than other methods when two batches are being aligned. When challenged with more than two batches, it failed completely (bottom left panel of **Fig. 2a**).

We believe that our adversarial batch correction strategy is efficient and scalable. Firstly, as demonstrated in the last response letter, an additional batch discriminator NN induces only a small parameter burden (< 1% of all trainable parameters). Secondly, the additional batch discriminator NNs do not complicate the training procedure, which remains to be two adversarial steps per iteration regardless of how many batch vectors are used (see Line 412-416, Page 19 for detailed description). We did not observe training instability or noticeable slowdown in practice when using multiple batch discriminators. In fact, our adversarial batch correction strategy makes it trivial to scale to a large number of batches (Line 496-498, Page 22) and also multiple sources of batch effect. As demonstrated in our benchmark, our approach is more effective in eliminating batch effect compared to scSphere.

[§] scSphere and SAUCIE were also added in the dimension reduction benchmark (**Supplementary Fig. 3**).

Additional Figure 1 Performance of multilevel batch effect correction by scSphere. Cell embedding colored by (a) cell type (corresponding to **Supp. Fig. 5d**), (b) dataset (corresponding to **Supp. Fig. 5e**), (c) donor in “Baron_human” (corresponding to **Supp. Fig. 5f**), (d) donor in “Enge” (corresponding to **Supp. Fig. 5g**), (e) donor in “Muraro” (corresponding to **Supp. Fig. 5h**).

d) For benchmarking dimension reduction, a more proper way is to use the same number of dimensions, e.g., 10 (although for linear methods such as PCA need more PCs). Also, it's good to add the parameters for different methods as DM methods such as tSNE are sensitive to parameter settings.

Thanks for the suggestion. While it seems that using the same number of dimensions “fairer” in benchmarking, different dimension reduction methods have different suitable dimensionalities as pointed out correctly by the referee. We believe it makes more sense to compare different methods at their most suitable dimensionalities. Meanwhile, for the record, a comparison where the dimensionalities of all methods are fixed at 10 (except for tSNE, UMAP and SAUCIE where the dimensionality is 2) is shown in **Additional Fig. 2**.

Additional Figure 2 Dimension reduction benchmark with fixed dimensionality of 10. (a) Mean average precision of different dimensionality reduction methods in each of the seven benchmark datasets, obtained under 10 dimensions except for tSNE and UMAP where dimensionality is limited to 2 (corresponding to **Supp. Fig. 3b**). **(b)** Ranking methods by their mean average precision across multiple benchmark datasets (corresponding to **Supp. Fig. 3c**). Error bars indicate mean \pm s.d.

We agree that other hyperparameters apart from dimensionality may also influence performance. However, exhaustively traversing the entire hyperparameter space for each method is simply computationally infeasible:

Method	Major hyperparameters apart from dimensionality
PCA	N.A.
ZIFA	N.A.
ZINB WaVE	ϵ (regularization weight)
tSNE	Initial PCA dimension, perplexity, exaggeration factor
UMAP	Initial PCA dimension, n_neighbors, min_dist
Dhaka	NN architecture
scScope	NN architecture, T (recurrence depth)
DCA	NN architecture
scVI	NN architecture

Assuming “NN architecture” corresponds to just 2 hyperparameters (e.g., the number and dimensionalities of hidden layers), and 5 candidate values are tested for each hyperparameter, we would need to run $\frac{1+1+5+5^3+5^3+5^2+5^3+5^2+5^2}{9} \approx 50.8 \times$ the number of experiments in our current benchmark. Considering that our current dimension reduction benchmark cost about a week to finish on an HPC cluster using 500 CPUs, $50 \times$ experiments roughly translates into 1 year of computation time with equivalent hardware, which is unfortunately impractical. As such, we chose to use the default hyperparameter configuration of each method, which is the standard procedure employed by many benchmarking studies^{5,6,7}. Meanwhile, we provide all benchmarking pipelines in our Github repository (https://github.com/gao-lab/Cell_BLAST/tree/master/Evaluation). Users interested in certain hyperparameter choices or methods not currently included can easily modify the pipelines to conduct custom experiments.

e) The authors showed the Smart-seq2 data results in additional Fig.3. It's not clear the zero-inflation in Smart-seq2 data influence the results (Cell Blast used Gamma-Poisson distribution for the count data)?

Thanks for the helpful suggestion! We conducted a direct comparison for the performance of Cell BLAST using NB (negative binomial / Gamma-Poisson distribution) vs ZINB (zero-inflated negative binomial), when trained on Smart-seq2 and UMI-based data respectively (**Supplementary Fig. 2**). The performance difference between NB and ZINB is generally small ($\Delta\text{MAP} < 0.002$ in 10 out of 11 datasets), though it does seem that ZINB fits Smart-seq2 data slightly better than NB, while NB fits UMI-based data slightly better than ZINB, consistent with that observed in a recent correspondence⁸. On the other hand, partly due to the additional zero-inflation factor, ZINB is more likely to over-align data than NB,

risking false positive rate in terms of cell querying. E.g., when training model to align two datasets (“Quake_Smart-seq2_Fat” and “Quake_Smart-seq2_Brain_Non-Myeloid”, both are subsets of the “Quake_Smart-seq2” dataset used above), the ZINB-based model over-aligned B cells with oligodendrocytes, while the otherwise identical NB-based model did not (**Additional Fig. 3**). We believe that the risk of over-alignment outweighs the marginal gain in fitting plate-based non-UMI data, thus prefer the simpler NB as the generative distribution.

Additional Figure 3 ZINB-based models are more likely to over-align than NB-based ones. (a-b) “Quake_Smart-seq2_Fat” and “Quake_Smart-seq2_Brain_Non-Myeloid” aligned by an NB-based Cell BLAST model with $\lambda_b = 0.02$ (two times the default value), where cell embeddings are colored by (a) cell type and (b) dataset respectively. (c-d) “Quake_Smart-seq2_Fat” and “Quake_Smart-seq2_Brain_Non-Myeloid” aligned by an otherwise identical ZINB-based Cell BLAST model, where cell embeddings are colored by (c) cell type and (d) dataset respectively. The over-alignment between oligodendrocytes and B cells are highlighted in red circles. Apart from these cells, there seems to be another over-alignment between brain pericytes and mesenchymal stem cells, but this is consistent with the fact that these cell types are related and share a certain transcriptomic signature⁹.

f) Should doublets be removed first for using Cell blast, or cell blast is sensitive to annotate doublets?

Thanks for the insightful comment. For doublets that consist of cells from the same cell type, Cell BLAST would annotate them as the corresponding constituent cell type, since the

encoder network contains a normalization step (**Equation 6**) and increase in library size would not make any difference.

However, for doublets that consist of cells from different cell types, it is possible that Cell BLAST gives undetermined prediction due to the probably uneven mixture of constituent cells. Thus, we advise standard data quality control including removal of low quality cells and potential doublets (<https://cblast.gao-lab.org/FAQs#what-preprocessing-steps-are-necessary>), which can be achieved using established methods^{10, 11, 12, 13}.

Referee 2 Comments

I would like to thank the authors for adequately addressing all the comments. I believe the manuscript has improved significantly.

The only additional remark I have is that I think Supplementary Fig1A deserves to be in the main text to help the readers with the model interpretation. After all, this is a methods paper and I find it strange that all the method description goes to the supplementary materials.

Thanks for the encouraging comment! We included only the flowchart in the main text to quickly give readers an idea of the overall components and purposes of the Cell BLAST suite. As suggested, we have turned **Fig. 1** into an illustration-only figure by combining the flowchart and illustration of model architecture.

References

1. Baron M, *et al.* A Single-Cell Transcriptomic Map of the Human and Mouse Pancreas Reveals Inter- and Intra-cell Population Structure. *Cell Syst* **3**, 346-360 e344 (2016).
2. Ding J, Regev A. Deep generative model embedding of single-cell RNA-Seq profiles on hyperspheres and hyperbolic spaces. *bioRxiv*, 853457 (2019).
3. Korsunsky I, *et al.* Fast, sensitive and accurate integration of single-cell data with Harmony. *Nat Methods*, (2019).
4. Amodio M, *et al.* Exploring single-cell data with deep multitasking neural networks. *Nat Methods*, (2019).
5. Sun S, Zhu J, Ma Y, Zhou X. Accuracy, robustness and scalability of dimensionality reduction methods for single-cell RNA-seq analysis. *Genome Biol* **20**, 269 (2019).
6. Abdelaal T, *et al.* A comparison of automatic cell identification methods for single-cell RNA sequencing data. *Genome Biol* **20**, 194 (2019).
7. Tran HTN, *et al.* A benchmark of batch-effect correction methods for single-cell RNA sequencing data. *Genome Biol* **21**, 12 (2020).
8. Svensson V. Droplet scRNA-seq is not zero-inflated. *Nat Biotechnol* **38**, 147-150 (2020).
9. Crisan M, *et al.* A perivascular origin for mesenchymal stem cells in multiple human organs. *Cell Stem Cell* **3**, 301-313 (2008).

10. McGinnis CS, Murrow LM, Gartner ZJ. DoubletFinder: Doublet Detection in Single-Cell RNA Sequencing Data Using Artificial Nearest Neighbors. *Cell Syst* **8**, 329-337 e324 (2019).
11. DePasquale EAK, *et al.* DoubletDecon: Deconvoluting Doublets from Single-Cell RNA-Sequencing Data. *Cell Rep* **29**, 1718-1727 e1718 (2019).
12. Wolock SL, Lopez R, Klein AM. Scrublet: Computational Identification of Cell Doublets in Single-Cell Transcriptomic Data. *Cell Syst* **8**, 281-291 e289 (2019).
13. Bais AS, Kostka D. scds: computational annotation of doublets in single-cell RNA sequencing data. *Bioinformatics* **36**, 1150-1158 (2020).

Reviewers' Comments:

Reviewer #1:

Remarks to the Author:

I would like to thank the authors for responding comprehensively to my concerns. The additional analysis considerably strengthened the paper. I look forward to seeing a published version.

I only have three minor comments below:

1, For benchmarking, it's important to use the same inputs for different methods. For example, for methods requiring counts as input, the same set of genes should be used (I think the authors did that, but it's good to write this down in the paper). Also, it should be helpful for the authors to write down the number of genes used or the number of PCs used. The authors can put such information in the figure legends or a table.

2, For dimension reduction benchmark (Supplementary Figure 3), I didn't see much difference as the error bars are highly overlapping except for SAUCIE (not sure why?). Detail parameter setting (maybe in a table) is of great help for readers interpreting the results. Similar to Figure 2a.

3, The authors claim 'Furthermore, it elegantly scales to a large number of batches' is weak as the authors did not show its scalability to a large number of batches. In the manuscript, the authors demonstrated that CELL-Blast worked for the case with several batches based on a single run on one dataset only, still lots of unknown about this approach in different scenarios, e.g., over-correcting with a large number of batches and cell type distributions in different batches are skewed.

Detailed Responses to Referees' Comments

Referee 1 Comments

I would like to thank the authors for responding comprehensively to my concerns. The additional analysis considerably strengthened the paper. I look forward to seeing a published version.

Thanks for the encouraging comment!

I only have three minor comments below:

1, For benchmarking, it's important to use the same inputs for different methods. For example, for methods requiring counts as input, the same set of genes should be used (I think the authors did that, but it's good to write this done in the paper). Also, it should be helpful for the authors to write down the number of genes used or the number of PCs used. The authors can put such information in the figure legends or a table.

Thanks for the insightful comments. We indeed used identical gene sets for all methods in the benchmarks and experiments. Sizes of these gene sets have been added in **Supplementary Table 2**. Specific statements have been added in the **Methods** section (Line 487 Page 19, Line 512 Page 20, Line 539 Page 21, Line 556 Page 21, Line 663 Page 25)*.

2, For dimension reduction benchmark (Supplementary Figure 3), I didn't see much difference as the error bars are highly overlapping except for SAUCIE (not sure why?). Detail parameter setting (maybe in a table) is of great help for readers interpreting the results. Similar to Figure 2a.

Thanks for the comment. The error bars in **Supplementary Fig. 3c** represent performance variation across seven benchmark datasets, which are indeed largely overlapping for most well-performing methods. In comparison, in **Supplementary Fig. 3b** where individual datasets are plotted separately, the error bars representing technical variation from different random initializations do not overlap. In other words, when specific datasets are considered,

* The only exception is the cell querying benchmark, where the competing querying methods come with their own specially designed feature selection method. To avoid arbitrarily compromising the performance of these competing methods (e.g., scmap is shown to achieve best performance with 500 input genes¹. Enforcing the same Seurat-based gene set as Cell BLAST would likely decrease its performance), each method used its own gene selection method as recommended by the original authors (Line 648 Page 24).

the difference among methods is significantly larger than technical variation, while no method consistently outperforms others across multiple datasets. We believe that several design issues of SAUCIE might have contributed to its observed poor performance in our benchmark, including its limitation to 2-dimensional embeddings, and its complex encoder and decoder architecture (3 hidden layers each, as compared to 1 in most other NN-based methods, which makes it considerably harder to train²).

As being described in the manuscript (Line 487 Page 19, Line 512 Page 20), all hyperparameters are left at their default values when conducting benchmark experiments.

3, The authors claim 'Furthermore, it elegantly scales to a large number of batches' is weak as the authors did not show its scalability to a large number of batches. In the manuscript, the authors demonstrated that CELL-Blast worked for the case with several batches based on a single run on one dataset only, still lots of unknown about this approach in different scenarios, e.g., over-correcting with a large number of batches and cell type distributions in different batches are skewed.

Thanks for the insightful comment! We have incorporated the referee's comments and revised the statement accordingly (Line 267 Page 10).

References

1. Kiselev VY, Yiu A, Hemberg M. scmap: projection of single-cell RNA-seq data across data sets. *Nat Methods* **15**, 359-362 (2018).
2. Hu Q, Greene CS. Parameter tuning is a key part of dimensionality reduction via deep variational autoencoders for single cell RNA transcriptomics. Preprint at <https://www.biorxiv.org/content/10.1101/385534v4> (2018).